# GROOT-2: Weakly Supervised Multimodal Instruction Following Agents

**Shaofei Cai**[1,2]* **Bowei Zhang**[3]* **Zihao Wang**[1,2] **Haowei Lin**[1,2]
**Xiaojian Ma**[5] **Anji Liu**[4] **Yitao Liang**[1]†

[1]Institute for Artificial Intelligence, Peking University
[2]School of Intelligence Science and Technology, Peking University
[3]School of Electronics Engineering and Computer Science, Peking University
[4]Computer Science Department, University of California, Los Angeles
[5]Beijing Institute for General Artificial Intelligence (BIGAI)

{caishaofei,zhangbowei,zhwang}@stu.pku.edu.cn, xiaojian.ma@ucla.edu
{linhaowei,yitaol}@pku.edu.cn, liuanji@cs.ucla.edu

## Abstract

Developing agents that can follow multimodal instructions remains a fundamental challenge in robotics and AI. Although large-scale pre-training on unlabeled datasets (no language instruction) has enabled agents to learn diverse behaviors, these agents often struggle with following instructions. While augmenting the dataset with instruction labels can mitigate this issue, acquiring such high-quality annotations at scale is impractical. To address this issue, we frame the problem as a semi-supervised learning task and introduce GROOT-2, a multimodal instructable agent trained using a novel approach that combines weak supervision with latent variable models. Our method consists of two key components: constrained self-imitating, which utilizes large amounts of unlabeled demonstrations to enable the policy to learn diverse behaviors, and human intention alignment, which uses a smaller set of labeled demonstrations to ensure the latent space reflects human intentions. GROOT-2's effectiveness is validated across four diverse environments, ranging from video games to robotic manipulation, demonstrating its robust multimodal instruction-following capabilities.

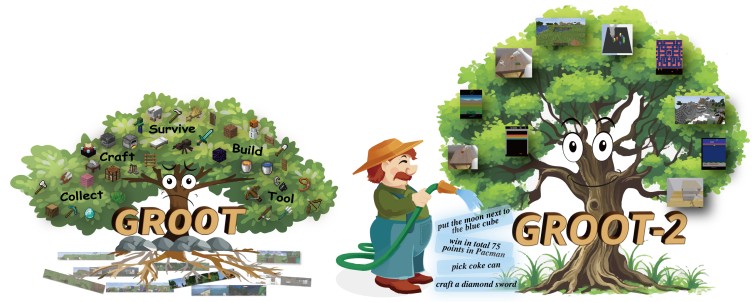

Figure 1: By feeding a mixture of demonstrations and some multimodal labels, we learn GROOT-2, a human-aligned agent capable of understanding multimodal instructions and adaptable to various environments, ranging from video games to robot manipulation, including Atari, Minecraft, Language Table, and Simpler Env.

## 1 Introduction

Developing policies that can follow multimodal instructions to solve open-ended tasks in open-world environments is a long-standing challenge in robotics and AI research. With the advancement of large-scale pretraining (Brown et al., 2020; Baker et al., 2022; Brohan et al., 2022), the research paradigm for instruction-following policies has shifted from reinforcement learning to supervised

---

*Equal contribution.    † Corresponding author.

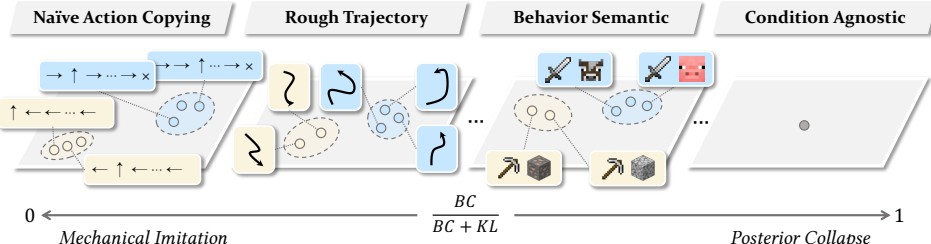

Figure 2: **The ELBO Objective of the VAE and Latent Space Spectrum.** We define a spectrum based on $R = \frac{BC}{BC+KL}$, where $R = 0$ corresponds to "mechanical imitation" and $R = 1$ to "posterior collapse." At low $R$, latent vector $z$ directly outputs action sequences without considering observations ($BC \to 0$). As $R$ increases, $z$ represents high-level task information, such as specific object interactions. At $R = 1$, $z$ provides no beneficial information for decision-making.

learning. In a supervised learning approach, researchers collect large amounts of demonstration data and annotate each demonstration with multimodal instructions—such as videos (Duan et al., 2017; Jang et al., 2022), texts (Padalkar et al., 2023; Lynch et al., 2023), and episode returns (Chen et al., 2021)—using hindsight relabeling. In theory, the instruction-following capability of such policies improves as the dataset grows. However, annotating demonstrations with high-quality multimodal labels is prohibitively expensive, making it challenging to scale these methods in practice.

Another line of work (Lynch et al., 2020c; Ajay et al., 2020; Cai et al., 2023b) avoids the need for additional human annotations by learning from demonstration-only data in a self-supervised manner. These approaches leverage latent variable generative models (Kingma & Welling, 2013) to jointly learn an encoder and a latent-conditioned policy. The resulting policy is capable of completing multiple tasks specified by a reference video (Cai et al., 2023b). While a reference video is generally expressive enough to represent various tasks, the inherent ambiguity in videos can lead to a learned latent space that is misaligned with human intention. For example, the encoder module may capture the dynamics between adjacent frames in a video, thereby learning a latent representation of the action sequence—a process we refer to as "mechanical imitation." While this latent space accurately reconstructs the target action sequence, the resulting latent representation is difficult for human users to leverage during policy deployment. Another potential issue is "posterior collapse," where the latent space collapses to a single point and loses its influence over the policy during inference. We attribute this mismatch between training and inference to the absence of direct supervision for aligning the latent space with human intention. As illustrated in Figure 2, an ideal controllable latent-induced policy space must strike a balance between these two extremes.

We present GROOT-2 (refer to Figure 1), a multimodal instructable agent developed using a latent variable model under weak supervision. To unify the training pipeline, we encode instructions from all modalities as distributions over the latent space. The training objectives consist of two key components: (1) **constrained self-imitating**, which utilizes large amounts of unlabeled demonstrations to enable the latent-conditioned policy to learn diverse behaviors; and (2) **human intention alignment**, which uses relatively small sets of multimodal labels to align the latent space with human intentions. Specifically, we apply the maximum log-likelihood method in the latent space for alignment. The underlying principle is that the latent embedding encoded by multimodal labels should also be sampled from the distribution learned from the corresponding video. Our approach is both general and flexible, as demonstrated through evaluations across four diverse environments—ranging from video games to robotic manipulation—including Atari Games (Bellemare et al., 2013), Minecraft (Johnson et al., 2016), Language Table Lynch et al. (2023), and Simpler Env (Li et al., 2024). These experiments highlight GROOT-2 's robust ability to follow multimodal instructions, with extensive tests showing that scaling up either unlabeled or labeled demonstrations further enhances performance.

## 2 BACKGROUND AND PROBLEMS

### 2.1 LATENT VARIABLE MODELS ENABLE CONTROLLABLE BEHAVIOR GENERATION

In recent years, the GPT series (Radford, 2018; Radford et al., 2019; Brown et al., 2020) has demonstrated impressive capabilities in controllable text generation. Its success can be attributed to

self-supervised pretraining and the advantageous properties of natural language. A natural language paragraph contains rich dependencies between sentences. For instance, the title of an article sets the central theme for its body, and the response in a question-answer or dialogue is highly correlated with the preceding text. This characteristic enables large language models, trained via next-token prediction, to achieve controllable text generation through prompts during inference. Unfortunately, such strong correlations do not exist between low-level actions. A desired behavior may not have a necessary preceding trajectory segment. Thus, it isn't easy to prompt a pre-trained policy model to generate a desired behavior. Instead, the generation of actions depends on an underlying latent intention variable. A natural approach is to employ latent variable generative models to jointly model trajectory data and the latent variables that drive them, allowing for controllable behavior generation by manipulating the latent variables during inference. Next, we will elaborate on how latent variable models model trajectory data.

As a classic latent variable generative model, Variational Autoencoder (VAE, Kingma & Welling (2013)) has been widely used in fields such as image generation and text generation. With the development of the offline pretraining paradigm, recent years have seen an increasing number of works utilizing VAE to model trajectory data. Typically, its architectures consist of three components: a posterior encoder, a prior encoder, and a policy decoder. The posterior encoder, $q(z|\tau)$, encodes a specific behavioral trajectory $\tau = (\mathbf{o}_{1:N}, \mathbf{a}_{1:N})$ and generates a posterior distribution over the latent space. When the action sequence can be accurately inferred from the observation sequence (Baker et al., 2022; Zhang et al., 2022)—i.e., when the inverse dynamics model of the environment $p_{\text{IDM}}(\mathbf{a}_{1:N}|\mathbf{o}_{1:N})$ is easily learned—the action sequence can be excluded from the posterior's input (Cai et al., 2023b), thus reducing the distribution condition to $\mathbf{o}_{1:N}$. The prior encoder, $p(z|\mathbf{o}_{1:k})$, generates a distribution over the latent space based on the history of observations, where $k$ denotes the length of the observation window. When $k = 0$, the prior distribution is independent of historical observations and is typically assumed to follow a standard normal distribution $\mathcal{N}(0; 1)$. The decoder, $\pi(\mathbf{a}_t|\mathbf{o}_{1:t}, z)$, is generally a latent-conditioned policy that takes in the environment's observations along with a specific latent variable to predict the next action to be executed. According to variational inference theory, we can optimize the VAE's modeling capabilities by maximizing the Evidence Lower Bound (ELBO)

$$\mathcal{L}_{\text{ELBO}} = \mathbb{E}_{z \sim q(z|\mathbf{o}_{1:N})} \left[ \sum_{t=k}^{N} -\log \pi(\mathbf{a}_t|\mathbf{o}_{1:t}, z) \right] + D_{\text{KL}}(q(z|\mathbf{o}_{1:N}) \parallel p(z|\mathbf{o}_{\leq k})). \quad (1)$$

There are generally three main objectives for using VAE to model trajectory data: (1) **Modeling multimodal behaviors** (Lynch et al., 2020a; Mees et al., 2022): For instance, when trajectory data is collected from different individuals, the variations in action sequences across different behavior modes can be substantial. Directly applying a naive behavior cloning algorithm may result in poor modeling performance. Introducing an additional latent variable to differentiate between behavior modes can help mitigate conflicts between them during training. (2) **Skill discovery** (Xu et al., 2023; Gupta et al., 2019): Complex trajectory data is often composed of various skills. A VAE can abstract action sequences in a self-supervised manner, enabling skill reuse in downstream tasks, such as accelerating the exploration process in reinforcement learning (Pertsch et al., 2021; Ajay et al., 2020). (3) **Following reference videos to complete open-ended tasks** (also known as one-shot demonstration learning, Cai et al. (2023b)): This approach aims to leverage the learned posterior encoder to recognize the underlying intention behind a reference video and encode it as a latent, which can then drive a policy to complete the specified task in a novel deployment. It points to a way to pre-train instruction-following policies using unlabeled trajectory data. We primarily focus on the third objective in the following paragraphs.

## 2.2 Modeling Behaviors with VAE Leads to Ambiguous Latent Space

Several studies on VAE (Alemi et al., 2018; Abeer et al., 2024) have pointed out that the Pareto frontier of the ELBO contains an infinite number of solutions for the latent space, a phenomenon we refer to as **latent space ambiguity**. To facilitate understanding, we provide an informal illustration in Figure 2, which shows several possible latent spaces when a VAE is used to model behaviors, all having similar ELBO values. We differentiate these latent spaces using the ratio $R = \frac{BC}{BC+KL}$, where $R = 0$ and $R = 1$ represent two extremes of the latent space. When $R$ approaches 0, the latent condition contains much information, nearly dictating every action of the policy's behavior. We

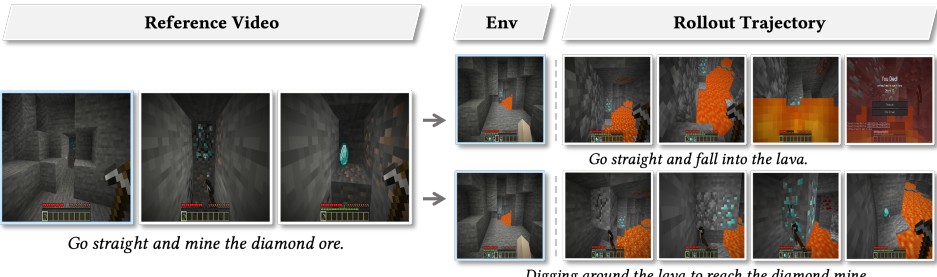

Figure 3: **Comparison of Policies with Different Latent Spaces.** The reference video depicts digging for diamonds. A policy that mechanically imitates the trajectory falls into lava, while one aligned with human intention avoids lava and successfully reaches the diamonds.

refer to this as **mechanical imitation**, where the VAE effectively degenerates into an Autoencoder (AE). Conversely, when $R$ approaches 1, the latent loses its ability to control the policy's output, a phenomenon known as **posterior collapse** (Fang et al., 2019; Pagnoni et al., 2018), in which the VAE reduces to an Auto-regressive (AR) model. Intuitively, as $R$ increases, the information encoded in the latent space becomes more high-level, and the policy relies more on environmental feedback (observations) to make decisions that align with the dataset's distribution. On the other hand, when $R$ is smaller, the policy tends to down-weight the environment's observations.

Not all latent spaces effectively support following a reference video. As shown in Figure 3, the gap between the environment state in the reference video and during policy deployment requires the posterior encoder to extract intentions independent of environmental dynamics. For instance, in the Minecraft task "mining a diamond underground," a reference video may show a player walking forward and mining a diamond. If the latent encodes only the trajectory sketch, the policy might fail by colliding with obstacles in the deployment environment. This mismatch occurs because humans interpret the video as "mining the diamond" rather than copying specific actions. Aligning the latent space with human intentions is critical for improving policy steerability.

## 3 ALIGNING POLICY LEARNERS WITH WEAK SUPERVISION

We explore the development of instructable agents based on latent variable models. To avoid "latent space ambiguity", we introduce human intention knowledge into the generative pretraining process of the policy model to assist in shaping the latent space. As multimodal labels associated with demonstrations carry rich human intention details, we propose a weakly supervised policy learning algorithm to leverage large amounts of unlabeled demonstration data to learn the latent space while using a small amount of multimodal labeled data to align the latent space with human intention. Ultimately, this enables instructions from all modalities to be unified within the same latent space. Next, we will elaborate on the dataset collection, training pipeline, and inference procedure.

**Dataset Collection.** We can collect two types of training data from the web: a large set of unlabeled demonstrations $\mathcal{D}_{\text{dem}} = \{(\mathbf{o}_{1:N}, \mathbf{a}_{1:N})\}$ and a relatively small set of annotated demonstrations $\mathcal{D}_{\text{lab}} = \{(\mathbf{o}_{1:N}, \mathbf{a}_{1:N}, \mathbf{w}_{1:M})\}$, where $\mathbf{o}$ is the image observation provided by the environment, $\mathbf{a}$ is the action taken by the policy, $\mathbf{w}$ is the word token, $N$ is the length of a demonstration, $M$ is the length of an annotation sentence. The annotation sentence can be multimodal, such as a language sentence (with $M \geq 1$) or a scaler of the episode return (with $M = 1$), which explains the behavior or outcome of the demonstration from a human's perspective. Since the annotation data is expensive to collect, we have $|\mathcal{D}_{\text{lab}}| \ll |\mathcal{D}_{\text{dem}}|$.

**Training Pipeline.** Our goal is to learn a shared latent space $\mathcal{Z}$, per-modal instruction encoders $e(z|c)$, and a latent-conditioned policy $\pi(\mathbf{a}_t|\mathbf{o}_{\leq t}, z)$. Leveraging past observations is essential for a policy to make decisions in a partially observable environment such as Minecraft (Johnson et al., 2016). We call the learned policy model GROOT-2, whose training pipeline is shown in Figure 4. For an unlabeled demonstration $(\mathbf{o}_{1:N}, \mathbf{a}_{1:N})$, we use the encoder module to produce a prior distribution $e(z|\mathbf{o}_1)$ and a posterior distribution $e(z|\mathbf{o}_{1:N})$. Using the reparameterization trick (Kingma & Welling, 2013), we sample the latent $z$ from the posterior distribution $e(z|\mathbf{o}_{1:N})$ and train the policy model, conditioned on $z$ and $\mathbf{o}_{1:t}$, to reconstruct the entire action sequence causally. To limit the information presented in

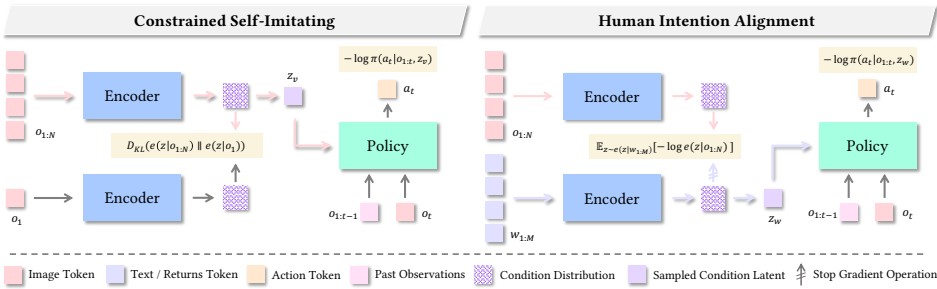

Figure 4: **Pipeline for Constructing a Training Batch for GROOT-2.** Each batch includes two sample types: (1) demonstration-only samples for learning a latent-conditioned policy (*Constrained Self-Imitating*); and (2) labeled samples (text or expected returns) for aligning the latent space with human intentions (*Human Intention Alignment*). The sample ratio varies by dataset distribution.

the latent space, we introduce an auxiliary KL divergence term in the objective:

$$\mathcal{L}_{\text{dem}}(\mathbf{o}, \mathbf{a}) = \mathbb{E}_{z \sim e(z|\mathbf{o}_{1:N})} \left[ \sum_{t=1}^{N} -\log \pi(\mathbf{a}_t|\mathbf{o}_{1:t}, z) \right] + \beta_1 D_{\text{KL}}(e(z|\mathbf{o}_{1:N}) \parallel e(z|\mathbf{o}_1)). \quad (2)$$

This allows the model to leverage demonstration-only data to enhance the complexity of the latent space, a process we refer to as "constrained self-imitating." For a labeled demonstration $(\mathbf{o}_{1:N}, \mathbf{a}_{1:N}, \mathbf{w}_{1:M})$, we pass the label $\mathbf{w}_{1:M}$ through the encoder module to obtain a latent distribution and train the policy model to reconstruct the action sequence based on the latent $z$ sampled from this distribution $e(z|\mathbf{w}_{1:M})$. This allows human knowledge to be modeled in the latent space. Further, to make the encoder understand demonstration $\mathbf{o}_{1:N}$ just like humans, we introduce an auxiliary MLE term: maximize the log-likelihood of $e(z|\mathbf{o}_{1:N})$ given the latent $z$ sampled from $e(z|\mathbf{w}_{1:M})$. Unlike the prior behavior cloning term, the aligning term can be quickly calculated in closed form. This process is referred to as "human intention alignment":

$$\mathcal{L}_{\text{lab}}(\mathbf{o}, \mathbf{a}, \mathbf{w}) = \mathbb{E}_{z \sim e(z|\mathbf{w}_{1:M})} \left[ \sum_{t=1}^{N} -\log \pi(\mathbf{a}_t|\mathbf{o}_{1:t}, z) \right] - \beta_2 \mathbb{E}_{z \sim \text{sg}[e(z|\mathbf{w}_{1:M})]} \left[ \log e(z|\mathbf{o}_{1:N}) \right], \quad (3)$$

where $\text{sg}[\cdot]$ denotes stop gradient operation. The MLE-based alignment objective ensures that the latent sampled from the label-conditioned distribution $e(z|w_{1:M})$ can also be sampled from its video-conditioned distribution $e(z|o_{1:N})$. The final loss function combines the two objectives:

$$\mathcal{L}(\mathcal{D}_{\text{dem}}, \mathcal{D}_{\text{lab}}) = \mathbb{E}_{(\mathbf{o}, \mathbf{a}) \sim \mathcal{D}_{\text{dem}}} \left[ \mathcal{L}_{\text{dem}}(\mathbf{o}, \mathbf{a}) \right] + \mathbb{E}_{(\mathbf{o}, \mathbf{a}, \mathbf{w}) \sim \mathcal{D}_{\text{lab}}} \left[ \mathcal{L}_{\text{lab}}(\mathbf{o}, \mathbf{a}, \mathbf{w}) \right]. \quad (4)$$

Specific implementation details, such as the model design choice, can be found in the Appendix A.

**Inference Procedure.** GROOT-2 supports two types of instructions during inference: (1) visual-based instruction – the user can either retrieve a demonstration from the dataset as a reference video or manually record a reference video to serve as the condition for the policy; (2) label-based instruction – the user can input a text sentence or specify an expected return as the condition (depending on the label modality used during the model's training). We tested them in the following experiments.

## 4 CAPABILITIES AND ANALYSIS

We aim to address the following questions: (1) How does GROOT-2 perform in open-world video games and robotic manipulation? (2) Can GROOT-2 follow instructions beyond language and video? (3) What insights can be gained from visualizing the learned latent space? (4) How does GROOT-2 scale with labeled and unlabeled trajectories? (5) What is the impact of backbone initialization on performance? (6) How do language and video losses influence performance?

**Environment and Benchmarks.** We conduct experiments across four types of representative environments: classical 2D game-playing benchmarks on Atari (Bellemare et al., 2013), 3D open-world gameplaying benchmarks on Minecraft (Johnson et al., 2016; Lin et al., 2023), and Robotics benchmarks on Language Table simulator (Lynch et al., 2023) and Simpler Env simulator(Li et al.,

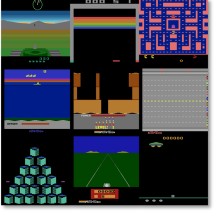 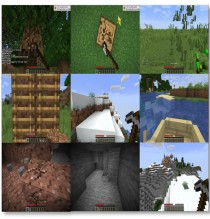 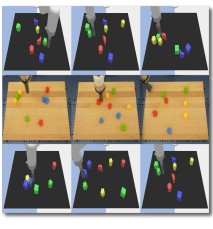 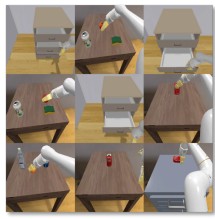

**(a)** Atari      **(b)** Minecraft      **(c)** Language Table      **(d)** Simpler Env

Figure 5: **Diverse visual environments used in the experiments.** We test our GROOT-2 on both video games (simple Atari games and the complex Minecraft game) and robotic manipulation environments (Language Table and Simpler Env). Minecraft is a partially observable open-ended environment, while others are fully observable.

Table 1: **Results on the Open-World Minecraft Benchmark.** This benchmark includes 8 task families and 100 tasks. Each task is evaluated 30 times across three seeds, and the average success rate is calculated per task family. For example, Combat (16) indicates 16 tasks in the Combat family.

| Methods | Prompt | Combat (16) | Hunt (10) | Ride (4) | Breed (8) | Craft (20) | Mine (20) | Interact (10) | Plant (12) |
|---|---|---|---|---|---|---|---|---|---|
| VPT | N/A | $11_{\pm 3}$ | $20_{\pm 4}$ | $7_{\pm 2}$ | $2_{\pm 0}$ | $4_{\pm 1}$ | $7_{\pm 2}$ | $21_{\pm 6}$ | $22_{\pm 7}$ |
| STEVE-1 | lang | $12_{\pm 3}$ | $9_{\pm 2}$ | $\mathbf{54}_{\pm 8}$ | $4_{\pm 2}$ | $5_{\pm 2}$ | $6_{\pm 3}$ | $53_{\pm 9}$ | $33_{\pm 8}$ |
| STEVE-1 | visual | $15_{\pm 4}$ | $10_{\pm 3}$ | $38_{\pm 9}$ | $6_{\pm 2}$ | $6_{\pm 2}$ | $10_{\pm 4}$ | $40_{\pm 8}$ | $43_{\pm 7}$ |
| GROOT-1 | visual | $18_{\pm 5}$ | $28_{\pm 8}$ | $26_{\pm 6}$ | $12_{\pm 3}$ | $15_{\pm 4}$ | $22_{\pm 7}$ | $57_{\pm 8}$ | $75_{\pm 6}$ |
| GROOT-2 | lang | $\mathbf{40}_{\pm 7}$ | $43_{\pm 5}$ | $46_{\pm 6}$ | $\mathbf{22}_{\pm 6}$ | $18_{\pm 3}$ | $\mathbf{37}_{\pm 5}$ | $55_{\pm 4}$ | $75_{\pm 9}$ |
| GROOT-2 | visual | $37_{\pm 4}$ | $\mathbf{48}_{\pm 7}$ | $51_{\pm 4}$ | $20_{\pm 4}$ | $\mathbf{27}_{\pm 3}$ | $36_{\pm 7}$ | $\mathbf{63}_{\pm 6}$ | $\mathbf{77}_{\pm 7}$ |

2024), illustrated in Figure 5. These four simulators are used to evaluate whether GROOT-2 can be effectively steered by returns (Chen et al., 2021; Mnih et al., 2015), reference videos (Cai et al., 2023b; Jang et al., 2022), and textual instructions (Brohan et al., 2022; 2023).

**Results on the Open-World Minecraft Benchmark.** To evaluate policy models in Minecraft, we used the contractor dataset from Baker et al. (2022), containing 160M frames. According to the meta information, labeled trajectories account for approximately $35\%$ of the total data. We extended the Minecraft SkillForge Benchmark (Cai et al., 2023b) from 30 to 100 tasks, grouped into eight families: Combat, Hunt, Ride, Breed, Craft, Mine, Interact, and Plant. Details are in the Appendix C. We compared GROOT-2 with three baselines: (1) VPT (Baker et al., 2022), a foundational model trained on YouTube data via imitation learning, lacking instruction-following; (2) STEVE-1 (Lifshitz et al., 2023), which supports text and future image-conditioned instructions; and (3) GROOT-1 (Cai et al., 2023b), a self-supervised model using reference videos as instructions. Key findings from Table 1 are as follows: (1) GROOT-2 (visual) consistently outperforms GROOT-1 across all task categories, with particularly notable gains in mob interaction tasks like Combat and Hunt. Comparing trajectories on Hunt, GROOT-1 mechanically repeats "attack" actions, while GROOT-2 actively tracks objects, showing that text data enhances object-centric understanding and better aligns with human intentions. (2) GROOT-2 (text) performs similarly to GROOT-2 (visual) across most tasks, demonstrating that language and visual modalities share task knowledge. This enables the model to leverage both modalities for improved task completion. This highlights the advantage of combining multimodal data for better alignment with human intentions and improved policy performance.

**Results on the Language Table benchmark.** To assess GROOT-2's multimodal instruction following capabilities in the context of Robotic Table Manipulation, we utilized the Google Language Table as our testing platform and compared it with methods such as LAVA (Lynch et al., 2023), RT-1 (Brohan et al., 2022), GROOT-1 (Cai et al., 2023b). We utilize a dataset provided by Lynch et al. (2023) comprising 100M trajectories. We removed the text labels from half of the trajectories in the dataset, creating a $1:1$ ratio of labeled to unlabeled trajectories. Given that the Language Table environment comes with five task families, all of which are instructed solely through language, we curated five reference videos for each task with relatively clear intentions to evaluate the model's ability to comprehend video instructions. Detailed specifics are provided in the appendix D. The experimental results are shown in the Table 2. We observed that: (1) GROOT-2 leads by an absolute success rate of $4\%$ following text-based instructions compared, likely due to GROOT-2's more

Table 2: **Results on the Language Table Benchmark.** We reported success rates (in %) within 200 steps for each instruction modality, averaging over 250 rollouts. Results are averaged over 3 seeds with mean and stderr. "-" indicates missing data. The percentages in parentheses indicate the proportion of labels used.

| Task Family | BC-Zero | LAVA | RT-1 | GROOT-1 | GROOT-2 (50%) | | GROOT-2 (100%) | |
| | lang | lang | lang | visual | lang | visual | lang | visual |
|---|---|---|---|---|---|---|---|---|
| block to block | - | $90_{\pm 2}$ | - | $8_{\pm 2}$ | $84_{\pm 9}$ | $78_{\pm 9}$ | $86_{\pm 8}$ | $82_{\pm 7}$ |
| block to absolute loc | - | $72_{\pm 4}$ | - | $10_{\pm 3}$ | $70_{\pm 8}$ | $68_{\pm 8}$ | $\mathbf{76}_{\pm 6}$ | $70_{\pm 8}$ |
| block to block relative loc | - | $72_{\pm 3}$ | - | $4_{\pm 1}$ | $74_{\pm 9}$ | $64_{\pm 7}$ | $\mathbf{76}_{\pm 8}$ | $62_{\pm 6}$ |
| block to relative loc | - | $64_{\pm 4}$ | - | $8_{\pm 2}$ | $82_{\pm 5}$ | $78_{\pm 6}$ | $\mathbf{84}_{\pm 6}$ | $80_{\pm 4}$ |
| separate two blocks | - | $94_{\pm 2}$ | - | $12_{\pm 2}$ | $98_{\pm 1}$ | $96_{\pm 2}$ | $\mathbf{98}_{\pm 0}$ | $98_{\pm 0}$ |
| **Overall** | $72_{\pm 3}$ | $78_{\pm 4}$ | $74_{\pm 13}$ | $8_{\pm 2}$ | $82_{\pm 8}$ | $76_{\pm 7}$ | $\mathbf{84}_{\pm 6}$ | $78_{\pm 8}$ |

Table 3: **Results on the Simpler Env Benchmark.** We report the success rate (in %) of the video-instruction and language-instruction following for each model on 3 task families. The percentages in parentheses indicate the proportion of labels used.

| Methods | Prompt | Pick Coke Can | | | | Move Near | Open/Close Drawer | | |
| | | H-Pose | V-Pose | S-Pose | Avg | Avg | Open | Close | Avg |
|---|---|---|---|---|---|---|---|---|---|
| RT-1-X | lang | **57** | **20** | **70** | **49** | 32 | 7 | **52** | 29 |
| Octo-base | lang | 5 | 0 | 1 | 1 | 3 | 0 | 2 | 1 |
| GROOT-2 (50%) | visual | 42 | 18 | 52 | 37 | 35 | **29** | 30 | 30 |
| | lang | 52 | 20 | 50 | 41 | 42 | 27 | 33 | 30 |
| GROOT-2 (100%) | visual | 40 | 22 | 47 | 36 | 35 | 27 | 33 | 30 |
| | lang | 53 | **23** | 52 | 42 | **45** | 27 | 35 | **31** |

refined model architecture design. We mark the results of RT-2 in gray here, as it uses significantly more training data than ours. (2) The performance of GROOT-2 in following video instructions dropped by approximately 6% compared to text instructions, possibly due to the ambiguity of the reference videos, where a "block to block" type video could be interpreted as a "block to relative location" type task. (3) GROOT-1 struggled to understand the intentions conveyed by the reference videos. We observed that GROOT-1 imitated a reference video's trajectory sketch rather than their colors and shapes. This further underscores the importance of introducing language annotations for some trajectory data as a crucial method to align with human intentions.

**Results on the Simpler Env Benchmark.** We utilized the Simpler Env (Li et al., 2024) simulation of the Google Robot environment to evaluate the policy's capability in controlling complex robotic arms. GROOT-2 is trained on the OpenX dataset (Collaboration et al., 2023). We erased the text labels from half of the dataset's trajectories, achieving a 1:1 balance between labeled and unlabeled data. We evaluated three types of tasks: *Pick Coke Can*, *Move Near*, and *Open/Close Drawer*. Following Li et al. (2024); Brohan et al. (2023), we set up multiple variants for each task. For instance, the *Pick Coke Can* task involved three different poses for the Coke can; in the *Move Near* task, the layout and types of objects varied; and in the *Open/Close Drawer* task, the drawer had three layers from top to bottom. We compared GROOT-2 with baseline methods such as RT-1 (Brohan et al., 2022), and Octo (Octo Model Team et al., 2024). Among these, RT-1-X is an efficient language-conditioned transformer-based policy trained on the entire OpenX (Collaboration et al., 2023) dataset, which can be considered the performance boundary that GROOT-2 can achieve. As shown in Table 3, we found that GROOT-2 (lang) and GROOT-2 (visual) achieved comparable performance to the RT-1-X model across all three tasks. This indicates that our method retains language control capabilities and imbues the policy with equivalent visual instruction control abilities.

**Can GROOT-2 Follow Instructions Beyond Language and Video, Like Episode Returns?**

We evaluated GROOT-2 's steerability and performance on four Atari games (Breakout, Demon Attack, Hero, and Name This Game). Datasets from Agarwal et al. (2020), containing approximately 10M frames per game, were used. Episode returns were normalized to $\mu = 0, \sigma = 1$.

For training, we constructed a dataset with 30% labeled trajectories (returns) and 70% unlabeled data. Using this dataset, we trained GROOT (wsl) (weakly supervised learning). For comparison, GROOT (ssl) was trained on the same dataset without return labels in a fully self-supervised manner.

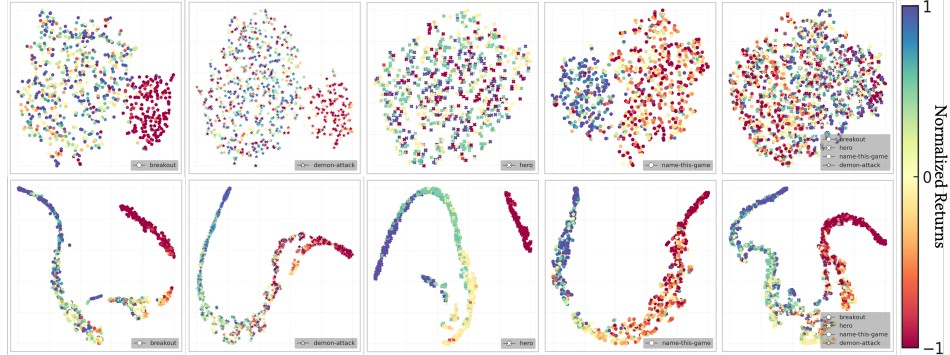

Figure 6: **Comparison of Weakly Supervised (WSL) and Self-Supervised (SSL) Learning on 4 Atari Games.** Policies are evaluated under return and reference video conditions. For return conditioning, normalized returns are input into the encoder, while for video conditioning, videos with similar returns (error $< 0.05$) are used.

Figure 7: **t-SNE Visualization of Learned Latent Spaces on Four Atari Games.** The first row shows results under self-supervised learning, while the second row displays GROOT-2 's performance under weakly supervised learning. Points represent reference videos, with shapes indicating games and colors denoting episode returns. The first four columns compare individual games, and the last column shows a mixed-game comparison.

Both models were jointly trained across the four games. During inference, we evaluated policy performance in following reference videos sampled from the test set with normalized returns $\{-1, 0, +1\}$, using 20 samples per category. Results (Figure 6) show: (1) GROOT (ssl) can recognize behavioral quality in reference videos, constructing a rough intention space even without labeled guidance. (2) Labeled data significantly improved GROOT (wsl)'s ability to understand video instructions, with the greatest gains in Hero and Name This Game. We also evaluated GROOT (wsl) on return-style instructions with normalized rewards $\{-1, 0, +1\}$. The similarity between video and reward-conditioned performance suggests the video encoder and reward encoder share the same intention space. The Atari experiments aim to evaluate GROOT-2 's performance on modalities beyond language and video, rather than maximizing scores, distinguishing it from traditional offline RL methods.

**What Does the Visualization of the Learned Latent Space Reveal?**

We applied t-SNE to visualize embeddings from randomly sampled reference videos. Each point in Figure 7 represents a unique video, with shapes denoting game environments and colors indicating episode returns. The first row illustrates results for GROOT (ssl), where videos in Breakout, Demon Attack, and Name This Game are classified into two categories based on episode return magnitudes, suggesting that the self-supervised algorithm distinguishes only significant score differences. In contrast, GROOT (ssl) shows poor clustering and limited steerability in the Hero game. The second row shows results for GROOT (wsl), which captures continuous variations in video behavior quality across all games. As shown in the fifth column, embeddings from different environments follow a continuous pattern aligned with reward labels, indicating a shared latent space that promotes cross-environment knowledge transfer.

**How Does Scaling Up Unlabeled Trajectories Impact Performance?**

We trained four GROOT-2 variants with $0\%$, $25\%$, $50\%$, and $100\%$ unlabeled data in Minecraft. Performance was tested on five Minecraft tasks (Chop Tree, Hunt Animals, Combat Enemies, Open Chest, Climb Mountain) and scored relative to skilled human players. For example, if a human collects 20.0 logs in 600 steps and GROOT-2 collects 15.0, the score is 0.75. Results (Figure 8) show consistent improvement with more unlabeled data, with the $100\%$ variant achieving a $5\times$ gain in the

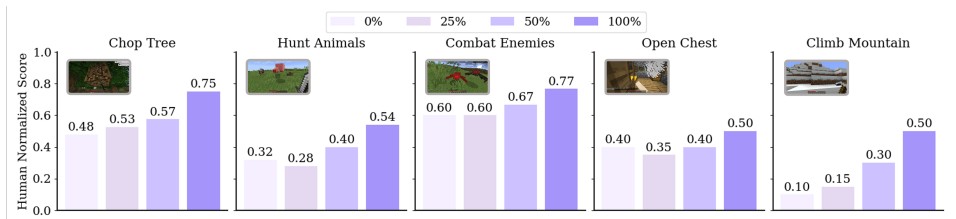

Figure 8: **Performance Comparison on Unlabeled Demonstrations.** Human-normalized task scores are averaged over 20 rollouts across 5 Minecraft tasks to evaluate the agent's reference-video following ability.

Climb Mountain task over the $0\%$ version. It is worth noting that the Climb Mountain and Open Chest tasks do not have language instructions in the training set.

### How Does Scaling Up Labeled Trajectories Impact Performance?

To evaluate the impact of labeled trajectory proportions in the training set on the instruction-following capabilities of GROOT-2, we conducted experiments on the Language Table benchmark. The total number of trajectories remained constant across different dataset configurations, with only the proportion of trajectories containing text labels varying. Figure 9 reports the success rate achieved by GROOT-2 conditioned on language. At low labeled data proportions $(0\% - 25\%)$, the success rate rapidly increased from $10\%$ to $65\%$, indicating that labeled data significantly influences model performance.

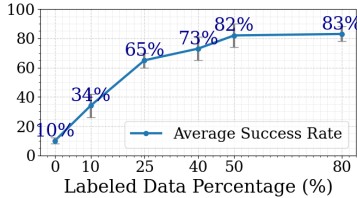

Figure 9: **Ablation Study on Labeled Trajectories in the Language Table.**

However, as the labeled data proportion increased to $50\% - 80\%$, the success rate plateaued, rising slightly from $82\%$ to $83\%$, demonstrating diminishing marginal gains from additional labeled data. Therefore, under resource constraints, a labeled data proportion of $50\%$ may represent the optimal balance between performance and cost.

### How Does Backbone Initialization Affect Performance?

We evaluated different initializations for ViT (random, ImageNet, CLIP) and BERT (random, BERT, CLIP) on the Language Table Benchmark. For randomly initialized models, both backbones were unfrozen during training. According to Table 4, CLIP initialization yielded the best results for ViT, followed by ImageNet, with minimal difference between them, while random initialization performed worst. For BERT, CLIP and standard BERT initialization performed similarly, both surpassing random initialization. Initializing vision and language encoders with CLIP parameters improves policy performance and reduces training time.

| Backbone | ViT | ViT | ViT/BERT | BERT | BERT |
|---|---|---|---|---|---|
| **Weights** | - | ImageNet | CLIP | - | BERT |
| **SR (in %)** | $76_{\pm10}$ | $80_{\pm11}$ | $\mathbf{82}_{\pm8}$ | $79_{\pm12}$ | $81_{\pm8}$ |

Table 4: Ablation study on the backbone initialization.

| Variants | $-\mathcal{L}_{\text{lab}}$ | baseline | $-\mathcal{L}_{\text{dem}}$ | baseline |
|---|---|---|---|---|
| **Prompt** | vision | vision | lang | lang |
| **SR (in %)** | $10_{\pm2}$ | $\mathbf{76}_{\pm7}$ | $12_{\pm3}$ | $\mathbf{82}_{\pm8}$ |

Table 5: Ablation on $\mathcal{L}_{\text{lab}}$ and $\mathcal{L}_{\text{dem}}$ objectives.

### How Does Language and Video Losses Impact Performance?

The $\mathcal{L}_{\text{lab}}$ loss significantly enhances the model's understanding of reference videos, as observed in the Language Table environment. We compared a variant without $\mathcal{L}_{\text{lab}}$ loss to the full GROOT-2 model, both trained on the same scale of the Language Table dataset, and tested their ability to follow reference videos using standard evaluation scripts. As shown in Table 5, the variant without $\mathcal{L}_{\text{lab}}$ loss failed to complete any tasks. Further analysis of its output videos revealed that it mechanically mimicked the arm movement trajectories in the reference videos, completely ignoring object colors and shapes, which is inconsistent with human understanding of the reference videos.

The $\mathcal{L}_{\text{dem}}$ loss is indispensable in the GROOT-2 architecture. Removing $\mathcal{L}_{\text{dem}}$ causes the pipeline to degrade into an autoencoder when processing unlabeled data. Without constraints on the latent encoding, the model tends to learn the video encoder as an inverse dynamics model, encoding low-level action sequences in latent z instead of high-level task information, thereby significantly reducing the behavior cloning loss. Additionally, Table 5 show that removing $\mathcal{L}_{\text{dem}}$ causes the language encoder's latent z to collapse, leading to a dramatic drop in task success rates.

## 5 RELATED WORKS

**Learning Policies Across Diverse Domains.** Developing policies for sequential control tasks in real and virtual environments poses significant challenges. Research spans domains such as robotic manipulation (Yu et al., 2019; Lynch et al., 2023), video games (Bellemare et al., 2013; Guss et al., 2019), and embodied navigation (Hong et al., 2020; Savva et al., 2019; Huang et al., 2023), with approaches categorized into reinforcement learning (RL) and imitation learning (IL) based on reward function reliance. For video games with dense rewards (e.g., ALE platform (Bellemare et al., 2013)), online RL algorithms can achieve superhuman performance (Mnih et al., 2015; Badia et al., 2020) but suffer from low efficiency, risky interactions, and limited generalization. These challenges restrict their applicability to physical (Padalkar et al., 2023) or embodied environments (Guss et al., 2019), where rewards and cheap interactions are unavailable. IL, as a supervised learning paradigm, addresses these issues through batch efficiency and scalability with large datasets, leveraging Transformer architectures (Zhang & Chai, 2021; Pashevich et al., 2021; Jang et al., 2022). The RT-X series (Brohan et al., 2022; 2023; Padalkar et al., 2023) advances robotic manipulation by training Transformers on large expert demonstration datasets, achieving strong zero-shot generalization. Similarly, Baker et al. (2022) developed a Transformer-based policy for Minecraft using internet-scale gameplay data, solving the diamond challenge. Building on this, Schmidhuber (2019) frames RL as supervised learning, while Chen et al. (2021); Lee et al. (2022) introduce "decision transformers" to model joint distributions of rewards, states, and actions from offline data, highlighting the potential for unified policy learning within Transformers.

**Learning Policies to Follow Instructions.** Enabling policies to follow instructions is key to building general-purpose agents. A common approach involves using language annotations from offline demonstrations to train language-conditioned policies (Abramson et al., 2020; Brohan et al., 2022; Reed et al., 2022; Cai et al., 2023a; Huang et al., 2023; Raad et al., 2024; Wang et al., 2023a;b), leveraging the compositionality of natural language for generalization. However, obtaining high-quality annotations is costly. An alternative uses anticipated outcomes as instructions. Majumdar et al. (2022) trained an image-goal conditioned navigation policy via hindsight relabeling (HER) (Andrychowicz et al., 2017) and aligned goal spaces with text. Similarly, Lifshitz et al. (2023) used this strategy for open-ended tasks in Minecraft. Generative latent variable models offer another solution, using label-free demonstrations to train plan-conditioned policies (Lynch et al., 2020b; Ajay et al., 2020). Extending this, Cai et al. (2023b) applied a posterior encoder to interpret reference videos in Minecraft. Policy learning with weak supervision remains less explored. Lynch & Sermanet (2020) proposed a shared latent space conditioned on language and HER-generated goal images, while Jang et al. (2022) replaced goal images with video labels under full supervision. Jain et al. (2024) trained robots using human videos as task representations but required extensive paired video-trajectory data. Myers et al. (2023) combined labeled and unlabeled trajectories, aligning start-goal pairs with language via contrastive learning, effective for Table Manipulation but limited in handling complex tasks or generalizing to partially observable environments like Minecraft.

## 6 CONCLUSIONS, LIMITATIONS AND FUTURE WORKS

This paper investigates the joint learning of a latent intention space and a multimodal instruction-following policy under weak supervision. We identify the "latent space ambiguity" issue in latent variable generative models when handling text-free trajectory data, arising from the absence of direct human guidance in shaping the latent space. To address this, we propose a weakly supervised algorithm for training GROOT-2. Evaluations across four diverse environments, from video games to robotic manipulation, demonstrate GROOT-2 's generality and flexibility in following multimodal instructions. However, GROOT-2 's reliance on trajectory data for training limits its applicability to video data, which lacks action labels. Considering the abundance and diversity of video data available online compared to trajectory data, extending the weak supervision framework to leverage both play and trajectory data would be a promising avenue for future work.

## 7 ACKNOWLEDGEMENTS

This work is funded in part by the National Key R&D Program of China #2022ZD0160301.

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

# A  IMPLEMENTATION DETAILS

## A.1  MODEL ARCHITECTURE

This section outlines the architectural design choices employed in our approach. GROOT-2 utilizes a Transformer encoder-decoder architecture, augmented with a probabilistic latent space. We detail the components of the model in a structured sequence: *extract representations*, *encode instructions*, and *decode actions*.

**Extract Representations.** This paragraph elaborates on the backbone networks used to extract representations from various data modalities. We denote the modalities of image observation, language instruction, and expected returns as $o_{1:N}$, $w_{1:M}$, and $r$, respectively. For vision inputs, we utilize a pre-trained Vision Transformer (ViT) (Dosovitskiy et al., 2020) initialized with CLIP (Radford et al., 2021) weights. Specifically, the $t-$step image observation $o_t$ is resized to $224 \times 224$ and processed to extract $7 \times 7$ patch embeddings $x_t^o = \left\langle x_{t,[1]}^o, \cdots, x_{t,[49]}^o \right\rangle$. The video representation $x^v$ is then composed of the averages of these embeddings across the video frames, denoted as $x^v = \langle \text{avg}(x_1^o), \cdots, \text{avg}(x_N^o) \rangle$, where $\text{avg}(\cdot)$ refers to spatial average pooling to minimize computational overhead and $N$ represents the video length. Textual inputs are processed using the BERT encoder (Devlin et al., 2019) of the CLIP model. Rather than utilizing the [CLS] token as the final representation, we retain all word embeddings generated by BERT as $x^w = \left\langle x_{[1]}^w, \cdots \right\rangle$. The BERT model parameters are kept frozen during training. For the scalar-form modality of expected returns, we employ a simple Multi-Layer Perceptron (MLP) to process these values, represented as $x^r \leftarrow \text{MLP}(r)$. These embeddings are then forwarded to subsequent modules.

**Encode Multimodal Instructions with Non-Causal Transformer.** Recent works (Reed et al., 2022; Lu et al., 2023; Team et al., 2023) have demonstrated the Transformer's effectiveness in capturing both intra-modal and inter-modal relationships, which inspires us to adopt a unified Transformer encoder for encoding multimodal instructions. This approach offers two significant advantages: (1) It eliminates the need for designing separate architectures and tuning hyperparameters for each modality. (2) It promotes the sharing of underlying representations across different modalities. Instructions are represented as a sequence of embeddings. Before encoding, each embedding is augmented with a modality-specific marker. For instance, video instructions are represented as $\langle x_1^v + [\text{VID}], \cdots, x_N^v + [\text{VID}] \rangle$, where $[\text{VID}]$ is a learnable embedding.

**Decode Action with Causal Transformer.** Given a latent $z$ and a temporal sequence of perceptual observations $o_{1:t}$, the policy aims to predict the next action $a_t$. Following prior works (Baker et al., 2022; Cai et al., 2023b; Raad et al., 2024), we employ the Transformer-XL model (Dai et al., 2019) in our policy network, which enables causal attention to past memory states and facilitates smooth predictions. Additionally, we utilize the shared vision backbone to extract vision representations, thereby representing perceptual inputs as $x_{1:t}^o$. A significant challenge with this approach is low efficiency: each new observation $x_t^o$ adds up to 49 tokens to the input sequence, substantially increasing memory and computational demands. To address this issue, we introduce a *pre-fusion mechanism* inspired by Abramson et al. (2020); Lynch et al. (2023); Alayrac et al. (2022). Specifically, we deploy a lightweight cross-attention module $\text{XATTN}(q = \cdot; kv = \cdot)$ to perform spatial pooling on $x_t^o$, using $z$ as the *query* and $\left\langle x_{t,[1]}^o, \cdots, x_{t,[49]}^o \right\rangle$ as the *keys* and *values*:

$$x_t^z \leftarrow \text{XATTN}(q = z; kv = x_{t,[1]}^o, \cdots, x_{t,[49]}^o). \tag{5}$$

This *pre-fusion mechanism* not only reduces the sequence length but also enhances the integration of perceptual and latent representations. Utilizing the latent-fused representations $x_{1:t}^z$ as the input sequence, we articulate the action decoding process in an *autoregressive* manner:

$$a_t \leftarrow \text{TransformerXL}(x_1^z, \cdots, x_t^z). \tag{6}$$

## A.2  HYPER-PARAMETERS

Hyper-parameters for training GROOT-2 are shown in Table 6.

Table 6: Hyperparameters for training GROOT-2.

| Hyperparameter | Value |
|---|---|
| Optimizer | AdamW |
| Weight Decay | 0.001 |
| Learning Rate | 0.0000181 |
| Warmup Steps | 2000 |
| Number of Workers | 4 |
| Parallel Strategy | ddp |
| Type of GPUs | NVIDIA A800 |
| Parallel GPUs | 8 |
| Accumulate Gradient Batches | 1 |
| Batch Size/GPU (Total) | 16 (128) |
| Training Precision | bf16 |
| Input Image Size | $224 \times 224$ |
| Visual Backbone | ViT/32 |
| Encoder Transformer | minGPT (w/o causal mask) |
| Decoder Transformer | TransformerXL |
| Number of Encoder Blocks | 8 |
| Number of Decoder Blocks | 4 |
| Hidden Dimension | 1024 |
| Trajectory Chunk size | 128 |
| Attention Memory Size | 256 |
| $\beta_1$ | 0.1 |
| $\beta_2$ | 0.1 |

## B  ATARI

**Environment Description.** Atari 2600 games contain a lot of diverse video games, which is a widespread benchmark to evaluate the decision-making capability of an agent. The Atari games do not inherently support multitasking concepts; agents are typically tasked with optimizing for the highest possible rewards. However, an advanced human player can deliberately control their gameplay level and achieve any potential score. The ability to "control scores" is generally considered a higher intelligence level compared with merely "winning the game". Therefore, this paper does not emphasize the highest absolute score an agent can achieve in the Atari environment. Instead, it focuses on evaluating the agent's ability to follow instructions in the form of videos and "desired cumulative rewards" and to perform at the appropriate level. Especially when videos serve as conditions, the agent needs to infer the player's level demonstrated in the reference gameplay, which poses a significant challenge for the current agents. To our knowledge, this setting has not been explored by previous works.

**Observation and Action Spaces.** We utilize the popular Arcade Learning Environment (ALE) as our testing platform, where the original observation image provided is $210 \times 160$, and the action space consists of 18 discrete actions defined by the joystick controller. Following previous works, the observation images are typically resized to $84 \times 84$ grayscale images. In implementing GROOT-v2, we employ the ViT/32 model initialized with OpenAI's pre-trained CLIP model. The observation image, 84x84, is resized to a resolution of $224 \times 224$ before being fed into the model for unification. During the training process, the ViT backbone is jointly fine-tuned. The TransformerXL architecture used for the decoder has its memory set to a horizon of 128.

**Training Dataset.** We utilize the trajectories from the Replay Buffer generated during the training of agents using the DQN algorithm on Atari, provided by Google, as our source of training data. We access this data through the interface provided in the d4rl-atari project at GitHub[*]. Specifically, the trajectory data for each game consists of three parts: mixed, medium, and expert, representing the environment interaction data from 0-1M steps, 9M-10M steps, and the final 1M steps of a training session, respectively. We construct training data of 10M steps for each Atari game, with the proportions of mixed, medium, and expert data being 2:5:3. During training, we use a single model to

---

[*]https://github.com/takuseno/d4rl-atari

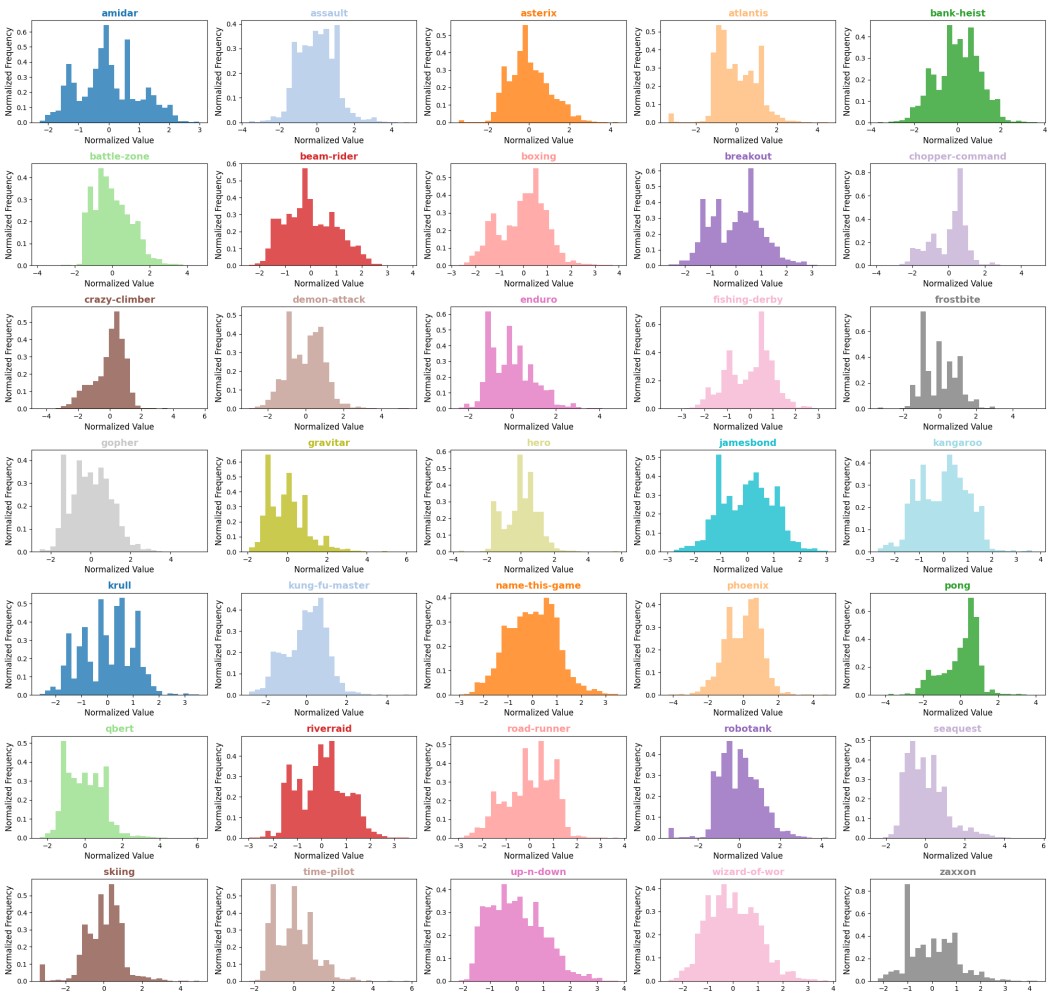

Figure B.1: **Distribution of episode returns for each Atari game.**

fit 35 selected game environments on Atari. Considering the significant differences in absolute scores across different games, we standardize the reward scores. Specifically, we calculate the cumulative reward scores for each complete trajectory and adjust them to a mean of 0 and a standard deviation of 1 using the formula $R \leftarrow (R - \mu)/\sigma$, which represents the game level corresponding to that trajectory. Figure B.1 illustrates the episode return distributions for each Atari game. Subsequently, each trajectory is segmented into 128-step fragments with the same label.

**Complete Results.** We selected trajectory data from 35 Atari games, totaling 350 million frames, to train GROOT-2, with 30% of the data labeled with returns and the remaining 70% containing only image observations and actions, aligning with the setup for weakly supervised training. After the model converged, we tested GROOT-2's ability to follow return-format and video-format instructions across these 35 games. When testing return-format instructions, we chose three samples within the normalized returns space: $\{-1, 0, 1\}$. For video-format instructions, we randomly sampled a segment of 128 frames from the test data with normalized rewards within the range of $\{-1, 0, 1\}$, allowing a deviation of $\pm 0.05$. Each instruction was tested 40 times, with the results depicted in Figure B.2. We observed the following: (1) In the majority of games, GROOT-2's performance showed a clear positive correlation with the game level corresponding to the instructions. (2) In certain games (such as Pong, Seaquest, Skiing, Wizard of Wor), video-format instructions yielded better control over the agent than return-format instructions. Conversely, in games like Amidar, Battle Zone, and Zaxxon, return-format instructions demonstrated significantly superior control compared to video-format instructions.

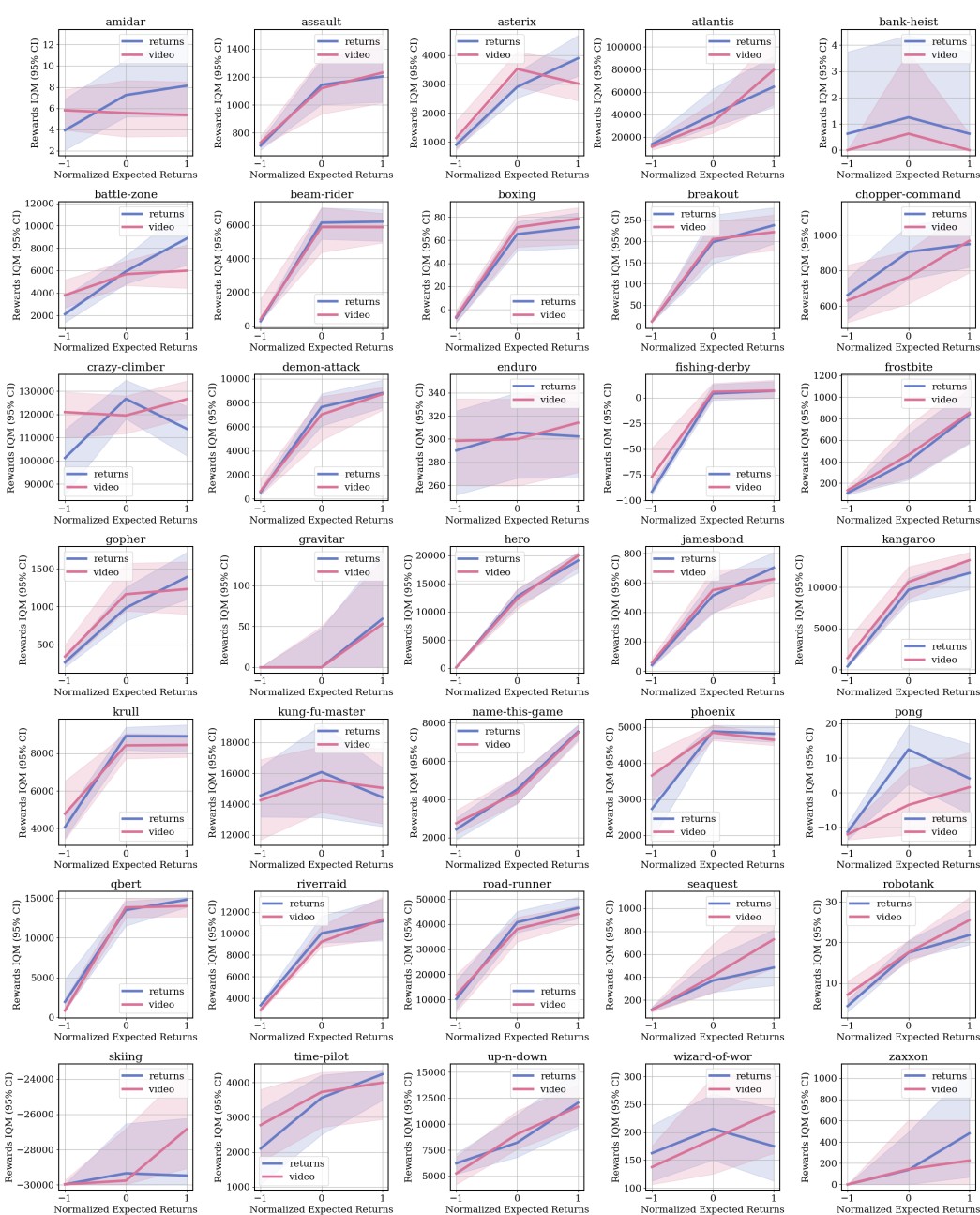

Figure B.2: **IQM scores (with** $95\%$ **confidence interval) of GROOT-2 which is jointly trained on 35 Atari games.** GROOT-2 can understand both the returns-format instructions and video-format instructions on most of the games. The performance of the GROOT-2 exhibits a positive correlation with the game level corresponding to the provided instructions.

# C  MINECRAFT

**Environment Description.** Minecraft is a 3D sandbox game with a global monthly active user base of 100 million. It features procedurally generated worlds of unlimited size and includes dozens of biomes such as plains, forests, jungles, and oceans. The game grants players a high degree of freedom to explore the entire world. The mainstream gameplay includes gathering materials, crafting items, constructing structures, farming land, engaging in combat mobs, and treasure hunting, among others. In this game, players need to face situations that are highly similar to the real world, making

judgments and decisions to deal with various environments and problems. One can easily specify a task with a natural language description or a demonstration video. Therefore, Minecraft is an ideal environment to test how an agent behaves in an open-world environment.

**Observation and Action Spaces.** We use the combination of 1.16.5 version MineRL (Guss et al., 2019) and MCP-Reborn[†] as our testing platform, which is consistent with the environment used by VPT (Baker et al., 2022) STEVE-1 (Lifshitz et al., 2023) and GROOT-1 (Cai et al., 2023b). Mainly because this platform preserves observation and action space that is consistent with human players to the fullest extent. On the one hand, this design brings about high challenges, as agents can only interact with the environment using low-level mouse and keyboard actions, and can only observe visual information like human players without any in-game privileged information. The Minecraft simulator first generates an RGB image with dimensions of $640 \times 360$ during the rendering process. Before inputting to the agent, we resize the image to $224 \times 224$ to enable the agent to see item icons in the inventory and important details in the environment. When the agent opens the GUI, the simulator also renders the mouse cursor normally. The RGB image is the only observation that the agent can obtain from the environment during inference. It is worth noting that to help the agent see more clearly in extremely dark environments, we have added a night vision effect for the agent, which increases the brightness of the environment during nighttime. Our action space is almost identical to that of humans, except for actions that involve inputting strings. It consists of two parts: the mouse and the keyboard. The mouse movement is responsible for changing the player's camera perspective and moving the cursor when the GUI is opened. The left and right buttons are responsible for attacking and using items. The keyboard is mainly responsible for controlling the agent's movement. To avoid predicting null action, we used the same joint hierarchical action space as Baker et al. (2022), which consists of button space and camera space. Button space encodes all combinations of keyboard operations and a flag indicating whether the mouse is used, resulting in a total of 8461 candidate actions. The camera space discretizes the range of one mouse movement into 121 actions. Therefore, the action head of the agent is a multi-classification network with 8461 dimensions and a multi-classification network with 121 dimensions.

**Training Dataset.** The contractor data is a Minecraft offline trajectory dataset provided by Baker et al. (2022), which is recorded by professional human players. In this dataset, human players play the game while the system records the image sequence $o_{1:N}$, action sequence $a_{1:N}$, and metadata $e_{1:N}$ generated by the players. Excluding frames containing empty actions, the dataset contains 1.6 billion frames with a duration of approximately 2000 hours. The metadata records the 7 kinds of events triggered by the agent in the game at each timestep, i.e. craft item, pickup, mine block, drop item, kill entity, use item, and custom. We augment each event with a text description using the OpenAI chatGPT service. To construct trajectory data with textual labels, we enumerate all timesteps within the trajectory where an event occurs. From this point, we count 112 frames backward and 16 frames forward to form a segment of 128 frames. The textual label for this segment is derived from the text associated with the event. It is important to note that many events occur frequently; for example, when the player is mining a tunnel, the event "mine block: cobblestone" is triggered on average twice per second. To address this issue, if a segment generated by an event overlaps with a previously generated segment, it is skipped. Each event collects a maximum of 2000 segments, and across all 1518 events, 414,387 segments are included. It is noteworthy that a significant amount of duplication persists within these segments, as a single segment may encompass multiple events.

## D  LANGUAGE TABLE

**Environment Description.** Language Table (Lynch et al., 2023) is a comprehensive evaluation suite proposed by the Google for assessing a robot's ability to follow natural language instructions to solve Table Manipulation tasks. It includes a dataset, environment, benchmarks, and a baseline policy. The evaluation benchmark encompasses over $87,000$ diverse behaviors and more than $600,000$ trajectories annotated with text instruction. In addition to data from real environments, the suite also provides a simulator akin to a real environment along with corresponding simulated data.

**Observation and Action Spaces.** Language Table's simulated environment resembles the real-world tabletop manipulation scenario, which consists of an xArm6 robot, constrained to move in a 2D plane with a cylindrical end-effector, in front of a smooth wooden board with a fixed set of 8 plastic blocks,

---

[†]https://github.com/Hexeption/MCP-Reborn

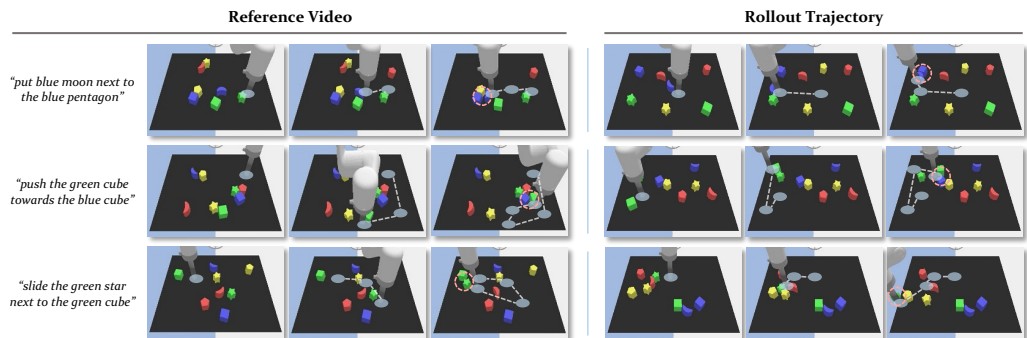

Figure C.3: **GROOT-2 can infer the intention behind the reference video and follow it to complete tasks.** The **left** visualizes three reference videos along with their textual descriptions. The right figure displays the policy's rollout trajectories when conditioned on the reference videos. The white dashed line represents the arm's movement trajectory, and the red dashed circle highlights the arm's final position.

comprising 4 colors and 6 shapes. In both simulation and real collection, they use high-rate human teleoperation with a 3rd person view (line-of-sight in real). Actions are 2D delta Cartesian setpoints, from the previous setpoint to the new one. They batch collected training and inference data to 5hz observations and actions.

**Training Dataset.** We use the training trajectories from the official Language Table repository. An oracle script generates the trajectories and covers all 5 task families, each containing 20M trajectories, in a total of 100M trajectories. The dataset names are: *language-table-blocktoblock-oracle-sim*, *language-table-blocktoblockrelative-oracle-sim*, *language-table-blocktoabsolute-oracle-sim*, *language-table-blocktorelative-oracle-sim*, *language-table-separate-oracle-sim*.

**Task Definition.** The evaluation benchmark consists of 5 task families (block2block, block2abs, block2rel, block2blockrel, separate), totaling 696 distinct task variants. We report the success rate of the agent within 200 steps on each task as the final metric. Considering that the Language Table inherently includes instructions in the language modality for its 5 task families, we have curated a set of reference videos for each task family, each with relatively clear intentions, to serve as a video instruction set. This is done to evaluate the model's ability to comprehend video instructions. The details are in Table 7. We visualize some examples when conditioning GROOT-2 on reference videos in Figure C.3.

## E  SIMPLER ENV

**Environment Description.** Simpler Env is a physical simulator proposed by Li et al. (2024), efficient, scalable, and informative complements to real-world evaluations. It can be used to evaluate diverse sets of rigid-body tasks (non-articulated / articulated objects, tabletop / non-tabletop tasks), with many intra-task variations (e.g., different object combinations; different object/robot positions and orientations), for each of two robot embodiments (Google Robot and WidowX).

**Observation and Action Spaces.** The observation and action spaces of Simpler Env are the same as the Language Table. The action sequence is expected to be a 6D end-effector pose trajectory with a gripper flag indicating the open/close status. Before feeding the image observation into the policy, we resize the image to a $224 \times 224$ resolution.

Table 7: Sampled reference videos to build video instruction set.

| Task Family | Video Description |
| --- | --- |
| block to block | put the red moon to the blue moon |
| block to block | put the blue moon towards the yellow star |
| block to block | slide the red pentagon close to the green cube |
| block to block | slide the green star to the red moon |
| block to block | put the green cube next to the red pentagon |
| block to absolute location | slide the blue cube to the upper left corner |
| block to absolute location | push the blue moon to the top left of the board |
| block to absolute location | move the red moon to the bottom left |
| block to absolute location | slide the yellow star to the right side of the board |
| block to absolute location | push the yellow pentagon to the left side |
| block to block relative location | move the green star to the left side of the yellow pentagon |
| block to block relative location | push the green star diagonally up and to the right of the green cube |
| block to block relative location | put the red moon to the bottom left side of the yellow star |
| block to block relative location | slide the yellow pentagon to the bottom left side of the red pentagon |
| block to block relative location | slide the blue cube to the top of the blue moon |
| block to relative location | push the green cube right |
| block to relative location | slide the yellow pentagon downwards and to the right |
| block to relative location | push the blue cube somewhat to the left |
| block to relative location | move the blue moon to the right |
| block to relative location | slide the red pentagon up |
| separate | pull the yellow pentagon apart from the blue moon |
| separate | pull the green star apart from the yellow star |
| separate | pull the blue cube apart from the blue moon and red pentagon |
| separate | move the blue cube away from the yellow star |
| separate | move the green star away from the yellow pentagon |

