# OpenReview forum: "GROOT-2: Weakly Supervised Multimodal Instruction Following Agents"
_ICLR.cc/2025/Conference — ICLR 2025 Poster_

### Official Review · Reviewer_Mcj1 · 2024-11-03

**Soundness:** 3
**Presentation:** 3
**Contribution:** 2
**Rating:** 6
**Confidence:** 3

**Summary:**

This paper introduces GROOT-2, a multimodal instruction-following agent trained with a novel weakly supervised learning framework. GROOT-2 addresses the challenge of multimodal instruction adherence in agents by reducing the reliance on labeled data, which is often costly and time-intensive to obtain. Instead, the model utilizes a two-part strategy: *constrained self-imitating* and *human intention alignment*. The constrained self-imitating method enables GROOT-2 to learn diverse behaviors from large sets of unlabeled demonstrations, while the human intention alignment component aligns the latent space of the model with human intentions using a smaller, labeled dataset. The authors validate GROOT-2’s performance across multiple environments, including video games like Atari and Minecraft, as well as robotic manipulation tasks, demonstrating that it outperforms previous models (e.g., GROOT-1 and STEVE-1) in multimodal instruction-following tasks.

**Strengths:**

1. **Novel Semi-Supervised Framework for Multimodal Instruction-Following:** The paper introduces a novel framework for training multimodal instruction-following agents with weak supervision. By framing the challenge as a semi-supervised problem, the authors effectively reduce reliance on costly labeled data, presenting a scalable alternative to fully supervised approaches. The combination of *constrained self-imitating* for leveraging large sets of unlabeled demonstrations and *human intention alignment* to ensure the latent space aligns with human intentions is both innovative and practical. This strategy addresses common issues like *posterior collapse* and *latent space ambiguity*, which have hindered multimodal instruction adherence in prior work. The proposed weakly supervised approach extends the applicability of instruction-following agents across a broad range of tasks and environments.

2. **Comprehensive Experimental Validation Across Diverse Environments:** The authors provide a rigorous empirical evaluation of GROOT-2 across multiple environments, including Atari games, Minecraft, and robotic manipulation tasks (e.g., Language Table and Simpler Env). The paper reports detailed quantitative comparisons against baseline models, such as GROOT-1 and STEVE-1, highlighting the model’s significant improvement across task families. Furthermore, the authors support the theoretical claims about their model with a thorough analysis of the learned latent space, illustrating how human intention alignment resolves ambiguities in latent representations. The experiments are comprehensive, with comparisons against varied baselines and multiple training data scales, showing the robustness and scalability of GROOT-2.

3. **Clear and Structured Presentation:** The paper is well-organized and clearly presents the problem, the proposed approach, and the experimental results.

4. **Broad Impact and Applicability of Weakly Supervised Instruction-Following:** The weakly supervised approach proposed by GROOT-2 has implications for the development of scalable, multimodal instruction-following agents. By reducing the reliance on labeled data, this framework broadens the feasibility of deploying such agents in real-world applications, where data annotation is often prohibitively expensive. The demonstrated effectiveness of GROOT-2 across diverse environments highlights its generalizability and adaptability. The contribution is particularly significant for applications in robotics, where the ability to follow human-like instructions across varied tasks without exhaustive labeling can accelerate the adoption of robotic agents.

**Weaknesses:**

1. **Scalability Limitations Due to Data Requirements**
   Although GROOT-2 successfully reduces the need for labeled data by using weak supervision, its current framework still relies on trajectory data, which includes both observations and actions. This dependence restricts the model’s scalability, as a vast amount of available data on the internet, such as play data or interaction data, often lacks explicit action information. The authors acknowledge this limitation but do not provide a concrete solution for incorporating such unlabeled play data. Addressing this issue could further expand the scalability of GROOT-2, potentially allowing it to leverage far larger datasets. Future work could explore methods for approximating or inferring actions from play data or combining GROOT-2’s framework with unsupervised or self-supervised approaches to handle action-less data.
**Recommendation:** Exploring methods to extend the framework to utilize play data would address a core scalability limitation.

2. **Evaluation Scope for Diverse Task Complexity:**  The paper demonstrates GROOT-2’s performance across varied environments; however, within each environment, the range of tasks presented may not fully capture the complexity of real-world, open-ended tasks. For instance, while Minecraft and Atari offer diverse scenarios, it remains unclear how well GROOT-2 would perform in environments with truly unpredictable dynamics or in tasks requiring complex, multi-step reasoning. An analysis of GROOT-2’s performance on tasks of varying complexity within each benchmark environment would provide a more granular understanding of the model’s adaptability and robustness. This could involve testing tasks with multiple dependencies or obstacles to simulate more realistic scenarios.
**Recommendation:** Evaluating GROOT-2 on tasks of varying difficulty would yield insights into the model’s robustness and real-world applicability.

**Questions:**

See above sections.

---

> ### Author Response · Authors · 2024-11-23
> **Response to Reviewer Mcj1**
>
> Thank you for your detailed and encouraging feedback. We are delighted that you found our semi-supervised framework for multimodal instruction-following innovative and practical, and that you appreciated its ability to address challenges like posterior collapse and latent space ambiguity. We are also grateful for your recognition of our comprehensive experiments across diverse environments and the clarity of the paper’s presentation. Your acknowledgment of the broad impact and applicability of our weakly supervised approach, particularly in reducing reliance on labeled data for real-world applications, is deeply motivating. We hope the following responses address your concerns and further clarify our contributions.
>
>
>
> **1. Scalability Limitations Due to Data Requirements**
>
> Thank you for raising such a professional and insightful question. Data relevant to training policies can be analyzed from two perspectives: (1) whether semantic-level labels, such as language or returns, are available; and (2) whether low-level labels, such as actions, are present. These two aspects are orthogonal and address different challenges.
>
> This study operates under the assumption that all training data contains low-level action labels. Our focus is on unifying the learning process for trajectories with and without semantic labels, using behavior cloning to provide the supervisory signal.
>
> However, in reality, most internet-sourced data consists of video data that lacks both semantic and action labels. Scaling up to such data would require addressing the absence of action labels. While this is beyond the scope of our current work, we are happy to share our perspective on this topic and welcome discussion with the reviewers. Broadly, we believe there are three potential pathways to address this issue:
>
> - *Leveraging the Inverse Dynamics Model:*
>
> Inspired by VPT [1, 2], an inverse dynamics model (IDM) - $p(a_t | s\_{t}, s\_{t+1})$ can be trained to label actions for adjacent frames in the video. Once trained, the IDM can be used to automatically annotate actions for all video data, enabling policy training at scale.
>
> - *Semi-Supervised Learning for Actions:*
>
> This approach involves splitting the data into two parts: (a) For a small subset of trajectory data with action labels, use behavior cloning as the optimization objective. (b) For the majority of video data without action labels, use next-frame prediction as the supervisory signal.
>
> Both tasks can share most of the network backbone, with separate task-specific heads (e.g., action head for behavior cloning and observation head for next-frame prediction). Since predicting the next observation requires similar capabilities to predicting the next action, the difference between $\hat{o}\_{t+1}$  and the current frame $o_{t}$ can help infer the intermediate action $\hat{a}\_{t}$. Learning from next-frame prediction can therefore improve the policy’s ability to predict actions. Related works such as GR-1 [3] and GR-2 [4] demonstrate how video prediction techniques can contribute to policy learning.
>
> - *Using a Latent Action Space:*
>
> As described in Genie [5], a latent action space can be defined and jointly learned with dynamic models and world models. This approach bypasses the need for action labels, optimizing the entire process end-to-end through next-frame prediction. Subsequently, the latent action space can be mapped to the real action space by fine-tuning the policy on a small dataset with action-labeled trajectories.
>
> It is worth emphasizing that the methods above represent a fully orthogonal direction for scaling up policies. We believe these techniques could be combined with the approach proposed in our paper, but such integration is beyond the scope of this study and is left as future work.
>
> *References:*
>
> [1] Video Pretraining (VPT): Learning to Act by Watching Unlabeled Online Videos
>
> [2] Learning to Drive by Watching YouTube Videos: Action-Conditioned Contrastive Policy Pretraining
>
> [3] Unleashing Large-Scale Video Generative Pre-Training for Visual Robot Manipulation
>
> [4] GR-2: A Generative Video-Language-Action Model with Web-Scale Knowledge for Robot Manipulation
>
> [5] Genie: Generative Interactive Environments

---

> > ### Author Response · Authors · 2024-11-23
> > **Response to Reviewer Mcj1 - Part 2**
> >
> > **2. Evaluation Scope for Diverse Task Complexity**
> >
> > In the field of embodied decision-making, completing a complex long-term task involves challenges beyond precise control of individual actions. A key aspect is the ability to plan and decompose the task into manageable sub-tasks. Task planning and decomposition require extensive commonsense knowledge and advanced reasoning capabilities, which, to date, have only been demonstrated effectively by large language models (LLMs) trained on massive language datasets. Consequently, a popular solution involves using LLMs to break down complex tasks into a series of short-term tasks and training a controller specifically to handle these short-term tasks.
> >
> > The GROOT-2 model proposed in this paper also functions as a controller, designed to acquire a repertoire of reusable skills that can be invoked by a higher-level planner. As such, multi-step reasoning is not the focus of this study. Additionally, during training, trajectories across all benchmarks were segmented into fragments no longer than 128 steps. This segmentation not only reduces the computational resources required for training but also simplifies the learning process for the latent space. Shorter trajectories result in latents that are reused more frequently, which further enhances their utility.
> >
> >
> >
> > **Thank you for your thoughtful feedback and for highlighting the strengths and context of our work. We hope this explanation clarifies the scope and focus of our study and addresses your concerns. If this resolves the issues raised, we kindly ask for your consideration in improving the score. Thank you again for your valuable input and constructive review!**

---

> ### Comment · Reviewer_Mcj1 · 2024-11-24
>
> Thanks for authors' detailed explanation! I'll keep my original score.

---

> > ### Author Response · Authors · 2024-11-25
> >
> > We sincerely appreciate your positive feedback and support. Thank you once again for your time and effort in reviewing our paper!

---

### Official Review · Reviewer_2co2 · 2024-11-03

**Soundness:** 2
**Presentation:** 3
**Contribution:** 2
**Rating:** 5
**Confidence:** 4

**Summary:**

This submission introduces GROOT-2, a weakly self-supervised instruction-following agent. It is trained in two stages, with the first being pretrained on large web data and the second being fine-tuning with small amount of human annotated data. Experiments are conducted on three different benchmarks, ranging from video games to robotics. It demonstrates competitive performance against previous approaches.

**Strengths:**

- Good writing and illustrative examples, easy to follow.
- Sufficient evaluation on three benchmarks.
- Good analysis.

**Weaknesses:**

- The idea to train latent-space conditioned policy is not novel. Although authors claim that the introduced "constrained self-imitating" and "human intention alignment" help to balance two extremes on the Pareto frontier, no ablations are showed to verity that.
- How is human annotation defined is unclear. Under the embodied AI and robotics context, human annotations usually refer to actions either collected or labeled by human. But in this work, human annotation seems to refer to high-level language instructions.
- Following the previous point, if action labels are indeed required to train the policies, how does the proposed method scale up to domains beyond video games and robot pick-and-place, where action labels are scarce?
- Experiment results seem not impressive enough. For those Minecraft tasks, the overall success rates are pretty low (only two tasks "interact" and "plant" succeed more than half). For robotics tasks, only the simple table-top picking-and-placing is demonstrated.

**Questions:**

1. L208: How web data come with action labels?
2. What is the task horizon in each environment?
3. Can authors show more complicated tasks? E.g., robotics tasks beyond table-top pick-and-place or real-robot experiments.

---

> ### Author Response · Authors · 2024-11-23
> **Response to Reviewer 2co2**
>
> Thank you for your kind and encouraging feedback. We are pleased to hear that you found the writing and examples clear and easy to follow, appreciated the comprehensive evaluation across four benchmarks, and valued the analysis provided in the paper. Your positive comments motivate us to further refine our work and address the concerns you raised, which we aim to clarify in the following responses.
>
>
>
> **1. Novelty**
>
> While latent variable models have been widely used in decision-making tasks to model behavioral data, most existing works [1, 2] primarily focus on capturing the multi-modal nature of behaviors. For example, in a task such as “moving object A in the bottom-left corner next to object B in the top-right corner,” there are often multiple feasible trajectories (e.g., moving via the top or the bottom). In such cases, the latent variables in a generative latent-conditioned policy typically represent one specific trajectory, rather than being used to guide the policy to perform distinct tasks. In these approaches, the decoder functions as the policy after training, while the encoder is not utilized during inference.
>
> In contrast, our work explores the role of latent variable models in decision-making from a fundamentally different perspective. We aim for the latent space to encode short-term tasks or reusable skills. During inference, users can leverage the encoder to extract task semantics from a reference video and control the policy to perform various tasks. To achieve this, we propose a novel modification to the training objective of latent variable models, enabling them to simultaneously learn from both labeled trajectory data (with text annotations) and unlabeled trajectory data. This dual capability has not been explored in previous works.
>
> *References:*
>
> [1] Learning Latent Plans from Play
>
> [2] Opal: Offline Primitive Discovery for Accelerating Offline Reinforcement Learning
>
>
>
> **2. Adding Ablation Studies to Verify Proposed Modules "Constrained Self-Imitating" and "Human Intention Alignment"**
>
> Thank you for your valuable suggestions. The proposed *constrained self-imitating* component corresponds to the video loss $\mathcal{L}\_\text{dem}$, while the human intention alignment component corresponds to the language loss $\mathcal{L}\_\text{lab}$. Here, we provide a detailed analysis of the impact of both the language loss ($\mathcal{L}\_\text{lab}$) and the video loss ($\mathcal{L}\_\text{dem}$) on the model’s performance.
>
> | Variants            | w/o $\mathcal{L}\_\text{lab}$ | baseline   | w/o $\mathcal{L}\_\text{dem}$ | baseline   |
> | ------------------- | ----------------------- | ---------- | ----------------------- | ---------- |
> | Instruction         | vision                  | vision     | lang                    | lang       |
> | Success Rate (in %) | $10 \pm 2$              | $76 \pm 7$ | $12 \pm 3$              | $82 \pm 8$ |
>
> The $\mathcal{L}\_\text{lab}$ loss significantly enhances the model’s understanding of reference videos, as observed in the Language Table environment. We compared a variant without $\mathcal{L}\_\text{lab}$ loss to the full GROOT-2 model, both trained on the same scale of the Language Table dataset, and tested their ability to follow reference videos using standard evaluation scripts. As shown in Table, the variant without $\mathcal{L}\_\text{lab}$ loss failed to complete any tasks. Further analysis of its output videos revealed that it mechanically mimicked the arm movement trajectories in the reference videos, completely ignoring object colors and shapes, which is inconsistent with human understanding of the reference videos.
>
> The $\mathcal{L}\_\text{dem}$ loss is indispensable in the GROOT-2 architecture. Removing $\mathcal{L}\_\text{dem}$ causes the pipeline to degrade into an autoencoder when processing unlabeled data. Without constraints on the latent encoding, the model tends to learn the video encoder as an inverse dynamics model, encoding low-level action sequences in latent $z$ instead of high-level task information, thereby significantly reducing the behavior cloning loss. Additionally, Table show that removing $\mathcal{L}\_\text{dem}$ causes the language encoder’s latent $z$ to collapse, leading to a dramatic drop in task success rates.
>
> The additional experiments and analysis have now been incorporated into the revised manuscript for a more comprehensive explanation.

---

> ### Author Response · Authors · 2024-11-23
> **Response to Reviewer 2co2 - Part 2**
>
> **3. Ambiguity in the Definition of Human Annotations**
>
> Thank you for pointing out this. We will add a clear definition of *annotation* in a prominent part of the paper to avoid confusion.
>
> Indeed, the understanding and significance of annotations vary between the robotics field and open-world gaming. In robotics, trajectory data collected via teleoperation often comes with a pre-specified language instruction, meaning that every trajectory is naturally annotated with a language label. Additionally, trajectories obtained through teleoperation are usually accompanied by specific action labels, eliminating the need for unsupervised or weakly supervised methods. However, when demonstrations are sourced from online platforms, action labels and language labels may be missing.
>
> This issue becomes even more pronounced in open-world video games like Minecraft and Atari. In such cases, action labels can often be generated automatically in post-processing, but the absence of language labels poses a more significant challenge. The problems of missing language labels and missing action labels are two entirely orthogonal research directions. This paper focuses on addressing the challenge of training instruction-following policies in the absence of language labels. While addressing the absence of action labels is beyond the scope of this work, we briefly discuss potential solutions to this issue in the following response.
>
>
>
> **4. Extending to Domains Without Action Labels**
>
> Thank you for raising such a professional and insightful question. Data relevant to training policies can be analyzed from two perspectives: (1) whether semantic-level labels, such as language or returns, are available; and (2) whether low-level labels, such as actions, are present. These two aspects are orthogonal and address different challenges.
>
> This study operates under the assumption that all training data contains low-level action labels. Our focus is on unifying the learning process for trajectories with and without semantic labels, using behavior cloning to provide the supervisory signal.
>
> However, in reality, most internet-sourced data consists of video data that lacks both semantic and action labels. Scaling up to such data would require addressing the absence of action labels. While this is beyond the scope of our current work, we are happy to share our perspective on this topic and welcome discussion with the reviewers. Broadly, we believe there are three potential pathways to address this issue:
>
> - *Leveraging the Inverse Dynamics Model:*
>
> Inspired by VPT [1, 2], an inverse dynamics model (IDM) - $p(a_t | s\_{t}, s\_{t+1})$ can be trained to label actions for adjacent frames in the video. Once trained, the IDM can be used to automatically annotate actions for all video data, enabling policy training at scale.
>
> - *Semi-Supervised Learning for Actions:*
>
> This approach involves splitting the data into two parts: (a) For a small subset of trajectory data with action labels, use behavior cloning as the optimization objective. (b) For the majority of video data without action labels, use next-frame prediction as the supervisory signal.
>
> Both tasks can share most of the network backbone, with separate task-specific heads (e.g., action head for behavior cloning and observation head for next-frame prediction). Since predicting the next observation requires similar capabilities to predicting the next action, the difference between $\hat{o}\_{t+1}$  and the current frame $o\_{t}$ can help infer the intermediate action $\hat{a}\_{t}$. Learning from next-frame prediction can therefore improve the policy’s ability to predict actions. Related works such as GR-1 [3] and GR-2 [4] demonstrate how video prediction techniques can contribute to policy learning.
>
> - *Using a Latent Action Space:*
>
> As described in Genie [5], a latent action space can be defined and jointly learned with dynamic models and world models. This approach bypasses the need for action labels, optimizing the entire process end-to-end through next-frame prediction. Subsequently, the latent action space can be mapped to the real action space by fine-tuning the policy on a small dataset with action-labeled trajectories.
>
> It is worth emphasizing that the methods above represent a fully orthogonal direction for scaling up policies. We believe these techniques could be combined with the approach proposed in our paper, but such integration is beyond the scope of this study and is left as future work.
>
> *References:*
>
> [1] Video Pretraining (VPT): Learning to Act by Watching Unlabeled Online Videos
>
> [2] Learning to Drive by Watching YouTube Videos: Action-Conditioned Contrastive Policy Pretraining
>
> [3] Unleashing Large-Scale Video Generative Pre-Training for Visual Robot Manipulation
>
> [4] GR-2: A Generative Video-Language-Action Model with Web-Scale Knowledge for Robot Manipulation
>
> [5] Genie: Generative Interactive Environments

---

> > ### Author Response · Authors · 2024-11-23
> > **Response to Reviewer 2co2 - Part 3**
> >
> > **5. Results on the Minecraft Benchmark.**
> >
> > The Minecraft benchmark used in this study differs significantly in difficulty from the robotics-related Language Table and SimplerEnv benchmarks.
> >
> > In robotics tasks, each trajectory clearly represents a specific task, and the task types are relatively limited, primarily including actions such as *pick*, *place*, and *move*. Furthermore, robotic environments are fully observable, with minimal randomness during the robot arm’s movements.
> >
> > In contrast, the Minecraft trajectories we used were collected from players freely exploring the game. These trajectories are significantly longer and contain a diverse range of behaviors, such as mining, farming, building, hunting, and navigating obstacles. To facilitate training, each long trajectory was evenly divided into segments of 128 frames. From a task complexity perspective, Minecraft is an open-world environment, which makes it substantially more challenging than the typically used Language Table and SimplerEnv benchmarks.
> >
> > Moreover, from an interaction standpoint, Minecraft is partially observable, and its observation space consists of high-dimensional RGB inputs at a resolution of 640x360. For example, the *Combat* task set includes a challenging “Hunt Enderman” task, where the Enderman teleports to evade attacks and deals significant damage. Even human players struggle to achieve a 100% success rate. Similarly, in the “Hunt Sheep” task, the agent must traverse complex terrain in a partially observable environment and continuously track randomly moving sheep until it gets close enough to kill them.
> >
> > Given these factors, GROOT-2’s success rates in Minecraft tasks are quite competitive. For comparison, the current SOTA methods in Minecraft, such as STEVE-1 and GROOT-1, achieve success rates lower than GROOT-2 on these benchmarks.
> >
> >
> >
> > **6. Task Horizon in Each Environment**
> >
> > In the Language Table Benchmark, as stated in the appendix, we follow the benchmark’s default inference script, which considers a task successful if completed within 200 steps. For the SimplerEnv Benchmark, we also adhere to the original inference scripts. For *move-related* tasks, the maximum step limit is set to 80, while for *drawer manipulation* tasks, the limit is 113 steps. In the Atari environment, the maximum step limit during inference is set to 5000 steps. For Minecraft, the step limit for each task is capped at 600 steps. We hope this clarifies the evaluation criteria used across the different benchmarks. Thank you for pointing this out.
> >
> >
> >
> > **7. Evaluation Scope of Robotic Tasks**
> >
> > In robotic environments, the skills learned by the policy trained in this work are inherently limited by the original trajectory dataset. Since the trajectories in the Language Table dataset are primarily of the *Move* type, and the trajectories in the SimplerEnv Benchmark mainly involve *Pick & Place* objects and *Push & Pull Drawer* tasks, the policies trained with our method in these environments can only perform similar types of tasks.
> >
> > It is important to emphasize that the goal of this study is not to train a policy capable of solving highly complex robotic control tasks. Instead, our focus is on developing a weakly supervised learning approach that can leverage both semantically labeled and unlabeled trajectories to train instruction-following policies. The experiments on Table Manipulation tasks are primarily intended to demonstrate the generalizability of our method across different domains.
> >
> > If you are interested in evaluating the policy’s ability to solve more complex tasks, we recommend referring to our experiments in Minecraft. The human demonstrations in the Minecraft dataset exhibit highly diverse behaviors. From this dataset, we selected 100 tasks categorized into 8 groups. Each task category includes significantly different tasks, all of which require recognizing RGB information and outputting precise actions to complete successfully.
> >
> >
> >
> > **We hope this explanation addresses your concerns and clarifies our contributions. Please let us know if this resolves the issue, and we would greatly appreciate your consideration for improving the score. Thank you once again for your valuable feedback!**

---

> > > ### Comment · Reviewer_2co2 · 2024-11-24
> > >
> > > Thank you authors for the response and extra experiments. I think they partially addressed my concern. I'd like to maintain my original evaluation.

---

### Official Review · Reviewer_SgYT · 2024-11-04

**Soundness:** 3
**Presentation:** 3
**Contribution:** 3
**Rating:** 6
**Confidence:** 4

**Summary:**

The authors propose an imitation learning method for training policies that can follow multi-modal instructions (either language or a video demonstration) using data that is only partially labeled with language. Like a prior paper, GROOT, they train a VAE on the expert data to obtain an encoder that produces a latent given a reference video as well as a latent-conditioned policy that predicts actions given an observation history and the latent. In this way the latent space models the demonstrator's intentions, and the policy can be commanded with a reference video by querying a latent from the encoder. This paper extends GROOT to the case where some of the expert data is labeled with language instructions. The authors modify the encoder to additionally accept language, and they introduce a new loss term that aligns the latents produced by language with those produced by the corresponding reference videos. In this way the language labels assist in learning a more well-structured latent space. Experiments on Minecraft, Atari, SIMPLER, and Language Table demonstrate that the proposed method, GROOT-2, outperforms GROOT and other relevant baselines when commanded with language or reference videos.

**Strengths:**

The paper addresses an important problem since language annotations are not always easy to acquire for imitation learning datasets. The method presented is a novel modification of an existing method which adds a non-trivial new capability. The explanation of "latent space ambiguity" is a nice justification for the method. The experiments cover a wide range of simulated settings.

**Weaknesses:**

My main concerns are some issues with the evaluation and the presentation/writing.

**Evaluation**

- It seems a central question is whether language loss ($L_{lab}$) improves reference video following capabilities and whether the reference video loss ($L_{dem}$) improves the language following capabilities. The cleanest way to answer this question would be to train with the GROOT-2 architecture and try removing either one of these loss components but keeping dataset size the same. However, I don't see this experiment. The atari experiment only tries removing $L_{lab}$.
- For each evaluation setting, please report what proportion of the training data is labeled with language and whether there are language labels for the tasks that were evaluated.
-  Only table 2 has error bars. It'd be great to have them for all the results.
- Why are there dashes in Table 2? Are the numbers taken from the prior papers (and they didn't provide the per task breakdown)? If so, did their evaluation protocol match your evaluation protocol?
- In the t-SNE plots there should be a legend showing what the colors mean.
- Why was a human-normalized score used as a metric for the Minecraft experiments focused on scaling up unlabeled demos, but a different metric used in Table 1?
- This sentence is confusing: "This enhancement may be due to the limitations of our text-demonstration data collection method, which might not include text annotations related to the Climb Mountain task." If there are no text annotations for this task, then what was the 0% unlabeled data policy trained on?

**Presentation/Writing**

- It'd be nice to state the main questions the experiments are designed to answer at the beginning of the section.
- The main text should clarify whether the encoder is a single model or multiple models.
- In tables 1, 2, and 3 bold the best result for each task. This would make them easier to read.
- The related work should discuss the relationship between prior work and the proposed method rather than just listing prior work. Additional papers to discuss/cite include GRIF which attempts to align task representations from language and initial state-goal pairs [1] as well as another work that uses videos as a task specifications [2]
- In the conclusion, "play data" does not typically mean data without actions, so I would avoid using this term here.

**Comments**

- CALVIN [3] would be a good benchmark in which to test this method since it contains both labeled and unlabeled data.

[1] https://arxiv.org/abs/2307.00117

[2] https://arxiv.org/abs/2403.12943v2

[3] https://arxiv.org/pdf/2112.03227

**Questions:**

- It seems the prior $e(z | o_1)$ is only ever input the first observation. Shouldn't the prior receive all the observations up to the current timestep $t$ like in equation 1?
- In the SIMPLER evaluation, did you use Octo-1.5 or the original Octo model? The original model actually works better on SIMPLER.

---

> ### Author Response · Authors · 2024-11-23
> **Response to Reviewer SgYT**
>
> Thank you for recognizing the importance of our work and the novelty of our approach. We greatly appreciate your positive feedback on our method, justification, and the breadth of our experiments. We are grateful for your thoughtful comments and support for our work. In the following, we aim to address the concerns you raised and provide clarifications and additional insights to further improve the manuscript.
>
> **1. How Does Language Loss $\mathcal{L}\_\text{lab}$ and Video Loss $\mathcal{L}\_\text{dem}$ Impact Performance**
>
> We sincerely thank the reviewer for pointing out this important question. Here, we provide a detailed analysis of the impact of both the language loss ($\mathcal{L}\_\text{lab}$) and the reference video loss ($\mathcal{L}\_\text{dem}$) on the model’s performance.
>
> |      Variants       | w/o $\mathcal{L}\_{lab}$ |  baseline  | w/o $\mathcal{L}\_{dem}$ |  baseline  |
> | :-----------------: | :---------------------: | :--------: | :---------------------: | :--------: |
> |     Instruction     |         vision          |   vision   |          lang           |    lang    |
> | Success Rate (in %) |       $10 \pm 2$        | $76 \pm 7$ |       $12 \pm 3$        | $82 \pm 8$ |
>
> The $\mathcal{L}\_\text{lab}$ loss significantly enhances the model’s understanding of reference videos, as observed in the Language Table environment. We compared a variant without $\mathcal{L}\_\text{lab}$ loss to the full GROOT-2 model, both trained on the same scale of the Language Table dataset, and tested their ability to follow reference videos using standard evaluation scripts. As shown in Table, the variant without $\mathcal{L}\_\text{lab}$ loss failed to complete any tasks. Further analysis of its output videos revealed that it mechanically mimicked the arm movement trajectories in the reference videos, completely ignoring object colors and shapes, which is inconsistent with human understanding of the reference videos.
>
> The $\mathcal{L}\_\text{dem}$ loss is indispensable in the GROOT-2 architecture. Removing $\mathcal{L}\_\text{dem}$ causes the pipeline to degrade into an autoencoder when processing unlabeled data. Without constraints on the latent encoding, the model tends to learn the video encoder as an inverse dynamics model, encoding low-level action sequences in latent z instead of high-level task information, thereby significantly reducing the behavior cloning loss. Additionally, Table show that removing $\mathcal{L}\_\text{dem}$ causes the language encoder’s latent $z$ to collapse, leading to a dramatic drop in task success rates.
>
> The additional experiments and analysis have now been incorporated into the revised manuscript for a more comprehensive explanation.
>
> **2.  What Proportion of the Labeled Trajectories During the Training**
>
> The ratio of labeled to unlabeled trajectories is maintained at 1:1 in both the Language Table environment (Table 2) and the Simpler Env environment (Table 3). In Minecraft, labeled data accounts for approximately 35% of the total dataset. In the Minecraft environment, all tasks in the training set are accompanied by corresponding language instructions, except for *Open Chest* and *Climb Mountain*, which do not include language instructions.
>
> Thank you for pointing this out. We have clarified this in the relevant section of the revised manuscript.
>
> **3. Questions about  Error Bars**
>
> Thank you for your valuable feedback. We have added error bars to the Minecraft experiment results in Table 1 of the revised manuscript.
>
> For the SimplerEnv benchmark, the baseline data was directly taken from the original paper [1], which did not report error bars in its tables. Additionally, we evaluated our proposed method using the official SimplerEnv scripts, where the initial setup for each task environment is fixed, and the evaluation is repeated 80 times. To maintain consistency, we believe it is preferable not to include error bars in this table.
>
> We appreciate your suggestion and hope this clarification addresses your concerns.
>
> *Reference:*
>
> [1] *Evaluating Real-World Robot Manipulation Policies in Simulation*
>
> **4. Data Points in Table 2**
>
> As you anticipated, the task-specific success rates were not reported in the original paper; only the average success rate across the five task categories was provided. However, the evaluation protocol used in their work is consistent with ours, as both rely on the evaluation scripts natively provided by the Language Table Benchmark.
>
> **5. Provide Legends in the t-SNE Plot**
>
> Thank you for your feedback. We have clarified the t-SNE visualization by explicitly labeling the value ranges corresponding to the continuous color gradient, which represents the continuous variation in returns from the reference video. We believe the updated figure is now much clearer and more informative.

---

> > ### Author Response · Authors · 2024-11-23
> > **Response to Reviewer SgYT - Part 2**
> >
> > **6. Metric Used for Minecraft Experiments**
> >
> > In our main Minecraft experiments, we selected a wide range of tasks with a relatively large task set. To ensure consistent evaluation, we focused on tasks where success rates could be easily computed based on the simulator’s reward feedback. Conducting human playthroughs to manually assess success rates for all these tasks would be both costly and unnecessary.
> >
> > For the ablation study, we performed comparisons on five diverse tasks. Since these tasks differ significantly and the evaluation metrics are not unified, normalizing the results based on human performance helps make the charts clearer and more interpretable. Additionally, this approach incurs minimal additional cost given the smaller task set.
> >
> > **7. Condition of the Prior Distribution**
> >
> > The formulation presented in Equation 1 adopts a general representation, where $k$ is a hyperparameter. In our specific implementation, we set $k = 1$, which aligns with previous works that model trajectory data using VAE [1, 2]. In our method, the prior distribution primarily serves a regularization role, and small variations in the hyperparameter k have minimal impact on the final performance.
> >
> > *References:*
> >
> > [1] Learning Latent Plans from Play
> >
> > [2] Opal: Offline Primitive Discovery for Accelerating Offline Reinforcement Learning
> >
> > **8. Missing Related Works**
> >
> > Thank you for pointing out the two related works we missed: **GRIF** and **Vid2Robot**.
> >
> > Similar to our work, GRIF focuses on leveraging both labeled and unlabeled trajectory data. It represents tasks in unlabeled trajectories using (start state, goal state) pairs and explicitly aligns these representations with language representations through contrastive learning. While this approach performs well in Table Manipulation tasks, the (start state, goal state) pair representation struggles to express complex, procedural task information and cannot generalize to partially observable environments (e.g., Minecraft). In such environments, the changes between the first and last frames are often insufficient to capture the full task context.
> >
> > Like our work, Vid2Robot uses video as a task representation. However, its training data consists of video-robot trajectory pairs, where each robot trajectory corresponds to a human video. During training, the human video serves as the task representation. This approach imposes higher data collection costs, as having humans demonstrate tasks in the environment is significantly more expensive than annotating videos with textual descriptions. In contrast, our method uses video and robot state sequences as task representations, avoiding the need for semantically matched video pairs by designing a weakly supervised learning approach.
> >
> > We have appropriately cited these works in the *Related Works* section of the revised manuscript. Thank you again for bringing this to our attention.
> >
> > **9. Additional CALVIN Benchmark**
> >
> > Thank you for bringing the CALVIN Benchmark to our attention. However, it appears that this benchmark has significant overlap with the capabilities demonstrated by the Language Table and SimplerEnv benchmarks used in our work, as all primarily focus on table manipulation tasks. Due to limited computational resources during the rebuttal phase, we are unable to incorporate experiments on this benchmark at this time. We will, however, consider utilizing the CALVIN Benchmark in future related work. Thank you again for the valuable suggestion.

---

> > > ### Author Response · Authors · 2024-11-23
> > > **Response to Reviewer SgYT - Part 3**
> > >
> > > **10. Addressing Presentation and Writing Concerns**
> > >
> > > - Clarifying the Purpose of the Experiments:
> > >
> > > We have added an introductory outline at the beginning of the experiment section to clearly explain the purpose of the experiments.
> > >
> > > - Number of Encoders:
> > >
> > > Our approach uses three encoders: (1) A video encoder based on the ViT and minGPT architectures, responsible for extracting task information from reference videos. (2) A text encoder, primarily implemented using the BERT architecture. (3) A returns encoder, which is mainly composed of an MLP.
> > >
> > > - Highlighting Best Results in Tables:
> > >
> > > We have bolded the best experimental results in all relevant tables for better clarity and readability.
> > >
> > > - Ambiguity in Explaining Play Data:
> > >
> > > Thank you for pointing this out. We agree that the term “play data” in the Conclusion section was unclear. We have replaced it with “video data” for greater precision.
> > >
> > > - Ambiguity in Minecraft Experiment Analysis:
> > >
> > > As clarified in the Appendix, the Contractor dataset used in this work is derived from human players freely exploring the game. Event information, such as kill_entity: sheep or mine_block: oak_log, was recorded during trajectory capture. However, these events are limited and template-based. For instance, some player behaviors like climbing or swimming do not have corresponding event records, which prevents us from generating text labels for these actions. As a result, trajectories related to such behaviors were excluded when filtering unlabeled data. Consequently, the policy’s video encoder struggles to interpret reference videos associated with climbing-related tasks.
> > >
> > > - Octo Baseline:
> > >
> > > The Octo-related data reported in the paper was directly taken from the original SimplerEnv paper for consistency.
> > >
> > >
> > >
> > > **We hope this response clarifies the points you raised and provides a satisfactory explanation. Please feel free to share any additional feedback or let us know if there are further aspects we can address. Thank you again for your thoughtful review!**

---

> > > > ### Comment · Reviewer_SgYT · 2024-11-26
> > > > **A few more questions**
> > > >
> > > > Thank you for answering my questions. A few remaining concerns/questions:
> > > > - In Language Table and SIMPLER, GROOT-2 is only given access to the labels for half of the dataset. Are the language-conditioned baselines given labels for all of the dataset?
> > > > - In Tables 2 and 3 it would be good to include the result for GROOT-2 trained on all the available labels for Language Table and SIMPLER. This would cleanly show the upper bound for GROOT-2 by just changing the amount of data with labels while keeping the architecture the same.
> > > > - I'm still a bit confused about the results for "Open Chest" and "Climb Mountain" in Figure 8. If there are no language labels for these tasks, then the policy trained with 0% unlabeled data must not have seen any examples of the task (either labeled or unlabeled). Does this mean that a policy trained on only labeled data from other tasks can already achieve non-zero success rates on "Open Chest" and "Climb Mountain"? Also, are the success rates in Figure 8 obtained through language or reference video conditioning?

---

> > > > > ### Author Response · Authors · 2024-11-27
> > > > >
> > > > > + All language-conditioned baselines were trained using fully supervised learning, meaning they utilized the entire text dataset.
> > > > > + Thank you very much for your suggestion. **We have added experiments on the Language Table and SimplerEnv benchmarks using a fully-labeled dataset to train the GROOT-2 model, and we have indicated the proportion of labels used.** Our results show that adding more text labels to previously unlabeled trajectories in the training set leads to only marginal performance improvements. For a detailed ablation study on different proportions of labeled trajectories, please refer to Figure 9.
> > > > > + We apologize for any confusion caused. As noted in the caption of Figure 8, the evaluation measures the agent’s ability to follow a reference video. Under the 0% labeled trajectories setting, training trajectories were never explicitly labeled with text instructions related to “Climb Mountains” or “Open Chest.” However, it is possible that behaviors like overcoming obstacles, climbing steps, or interacting with objects still appear in segments of the trajectories due to the data splitting process.
> > > > > This stems from the fact that the data was recorded by human players first, and text labels were added later through a relabeling process, rather than being recorded based on pre-defined text instructions. As a result, we cannot strictly exclude behaviors unrelated to the text labels from the training data.
> > > > > Nonetheless, even for tasks without explicit text labels, the agent can still be prompted to perform them using reference videos—a capability that GROOT-1 has already demonstrated. Thank you for highlighting this point, and we hope this clarification helps address your concerns.
> > > > >
> > > > > Thank you once again for your thorough review of our rebuttal and for providing us with valuable feedback for improvement. If you have any further concerns, we would be more than happy to address them.

---

### Official Review · Reviewer_QHCB · 2024-11-05

**Soundness:** 2
**Presentation:** 3
**Contribution:** 2
**Rating:** 5
**Confidence:** 4

**Summary:**

The authors propose a simple VAE style architecture to combine a small amount of language/return annotated data with a large demonstration-only dataset to learn a downstream policy. The authors test the method on Atari, Robotics benchmarks (Simpler and Language Table) and MineCraft. The authors compare their work to several per domain baselines and report strong improvements in MineCraft. Robotics results show smaller/statistically insignificant gains.

**Strengths:**

Strengths:
1. The paper is well written and easy to understand
2. The theoretical underpinnings of the work are clearly described using the R ratio diagrams. This motivates the work well.
3. The experimental suite is large and the method seems to work on a variety of benchmarks.
4. The question of utilizing labeled and label-free demonstrations needs to be answered in order to move the needle on generalist agents.

**Weaknesses:**

While I enjoyed reading the paper I have several issues with the current version of the paper. It reads more like a technical report for promotion purposes, not scientific material. I enlist my qualms below:
1. **How much supervision?:** While the authors have the word “weakly supervised”, it's unclear how much supervision is actually being used. Are all tasks covered in the labeled demonstration dataset? Is there any amount of language generalization that occurs, for e.g., are any tasks solvable zero-shot? How good are the demonstrations? What if there was play data/suboptimal demonstrations in the unlabeled demonstration dataset? Would the method still work? None of these questions are answered and I think they should be added since the focus of the work is weak supervision.
2. **No ablations/experiments describing model architectures?:** How does initializing the model with something other than BERT work? What about vision inputs? No ablations on ViT architectures initializations are provided. Works like LIV (https://arxiv.org/abs/2306.00958), R3M etc are available as baselines?
3. **Returns is not language:** The choice of adding of Atari is odd - there are no language labels here. Incidentally, it is only here that details of the amount of labels are provided. What is the distribution of labels? Are they mostly high reward trajectories? If return conditioned imitation is the focus, then works like Multi-game Decision transformer, CQL and other offline methods should be added as baselines. The rest of the paper does not consider return conditioning at all, so this section feels out-of-place.
4. **Missing baseline.** A very necessary baseline, I believe, is missing here. What if you trained a classifier on all the labeled data, which conditioned on a video would generate the textual label for the video. This labeler could be subsequently used to pseudo label all the demonstration-only data. Finally, a language conditioned imitation learning agent could be trained on this pseudo-labeled data along with the labeled demonstrations. This I believe is a natural baseline to your work. This is missing.
5. **Baselines are uninformative:** In the robotics section, why compare with RT-1? It’s trained on a different data distribution and it has a different architecture? This comparison generates more questions than answers. A clean experiment here would be retraining an open source Octo/OpenVLA model on all the labeled data only, i.e., only utilizing your labeled dataset.
6. **Missing ablation on amount of labeled data:** While I appreciate that the method scales with unlabeled data, what happens when the amount of labeled data is varied? Afterall, this is the limiting resource for this work.
7. Table 1 is missing error bars.

**Questions:**

1. How was beta_1 and beta_2  tuned? How was the R ratio controlled empirically?
2. Does the model finetune easily? If one were to use the model for a new set of tasks, would this transfer be data efficient

---

> ### Author Response · Authors · 2024-11-23
> **Response to Reviewer QHCB**
>
> Thank you for your detailed and thoughtful feedback, and for the significant effort you put into reviewing our work. We are pleased to hear that you found the paper well-written and easy to follow, and that the theoretical foundation and motivation behind our approach were clearly conveyed through the  ratio diagrams. We also appreciate your recognition of the broad experimental suite and the importance of addressing the challenge of leveraging both labeled and label-free demonstrations to advance the development of generalist agents. We hope the following responses address your concerns and further clarify our contributions.
>
> **1. What Proportion of the Labeled Trajectories During the Training**
>
> The ratio of labeled to unlabeled trajectories is maintained at 1:1 in both the Language Table environment (Table 2) and the Simpler Env environment (Table 3). In Minecraft, labeled data accounts for approximately 35% of the total dataset. In the Minecraft environment, all tasks in the training set are accompanied by corresponding language instructions, except for *Open Chest* and *Climb Mountain*, which do not include language instructions. Thank you for pointing this out. We have clarified this in the relevant section of the revised manuscript.
>
> **2. Impact of Demonstration Quality on Performance**
>
> In table manipulation environments, such as Language Table and SimplerEnv, all trajectories are of high quality, as they are explicitly collected to complete specific instructions. However, in the Atari environment, the trajectory quality varies, determined by the cumulative rewards achieved during gameplay, also known as the episode return. Our experiments in the Atari environment demonstrate that the latent space, learned through the generative model architecture, effectively encodes information related to the quality of trajectory behaviors. By manipulating the latent space, we can indirectly influence the quality of the policy’s actions.
>
> When low-quality trajectories are present in the training data, it is important to incorporate a quality parameter as part of the instruction. This allows the model to distinguish between high-quality and low-quality behaviors. Conversely, if high- and low-quality trajectories are mixed without distinction, it can negatively impact performance. From a generative modeling perspective, when behaviors of varying quality are modeled under the same distribution for a given instruction, there is a chance that suboptimal behaviors may be sampled, which can degrade overall performance.
>
> **3. Generalization**
>
> During training on the Language Table benchmark, we intentionally removed all language instructions related to the *blue cube* object. The corresponding trajectories were provided to the model as unlabeled demonstrations. We then tested the model on language instructions such as “move blue block to xxx” and observed that the model could still complete these tasks with a high success rate (~70%). This demonstrates the model’s ability to exhibit a certain level of generalization in the language domain.
>
> Additionally, in Minecraft, we selected the *Climb Mountain* and *Open Chest* tasks. Based on the text instruction generation method from the Contractor Data, we ensured that no related textual instructions were present during training for these tasks. During testing, we provided a reference video and found that the model could complete these tasks with a success rate of approximately 50%, as shown in Figure 8. This highlights the model’s capacity for generalization even in the absence of textual training instructions.
>
> **4. How Does Backbone Initialization Affect Performance**
>
> |      Variants       |      ViT      |      ViT      |   ViT/BERT   |     BERT      |     BERT     |
> | :-----------------: | :-----------: | :-----------: | :----------: | :-----------: | :----------: |
> | Pretrained Weights  |    Random     |   ImageNet    |     CLIP     |    Random     |     BERT     |
> | Success Rate (in %) | $76_{\pm 10}$ | $80_{\pm 11}$ | $82_{\pm 8}$ | $79_{\pm 12}$ | $81_{\pm 8}$ |
>
> We evaluated different initializations for ViT (random, ImageNet, CLIP) and BERT (random, BERT, CLIP) on the Language Table Benchmark. For randomly initialized models, both backbones were unfrozen during training. According to Table, CLIP initialization yielded the best results for ViT, followed by ImageNet, with minimal difference between them, while random initialization performed worst. For BERT, CLIP and standard BERT initialization performed similarly, both surpassing random initialization.  Initializing vision and language encoders with CLIP parameters improves policy performance and reduces training time.
>
> Thank you for pointing this out. We have updated the revised version accordingly.

---

> > ### Author Response · Authors · 2024-11-23
> > **Response to Reviewer QHCB - Part 2**
> >
> > **5. Are LIV and R3M Methods Avaliable as Baselines**
> >
> > Thank you very much for pointing out the relevance of LIV and R3M. Both works focus on multimodal (vision and language) joint pretraining, first learning semantic-rich visual representations from large-scale embodied task video data. In the second stage, these representations are used to train either a language-conditioned policy or an image-goal-conditioned policy. This is indeed a promising research direction.
> >
> > However, it is important to emphasize that our work aims to explore and address challenges in modeling trajectory data using latent variable generative models. Such models have the unique capability to directly utilize unlabeled trajectory data and enable users to control the policy by sampling from the learned latent space—a particularly compelling feature. That said, we observed that unconstrained latent variable generative models often produce latent spaces that deviate from human understanding. To address this, we introduce a limited amount of textual data (reflecting human preferences and interpretations of trajectory data) to help shape the latent space. Our approach ultimately supports both *reference video-based instructions* and *language-based instructions*.
> >
> > We believe that the focus of our research is fundamentally different from pretraining-based approaches, making LIV and R3M unsuitable as baselines for our study.
> >
> > **6. Provide the Episode Return Distributions in Atari Training Dataset**
> >
> > The trajectories collected for the Atari Benchmark originate from the replay buffer of the DQN reinforcement learning algorithm. The quality of these trajectories is categorized into three levels: expert, medium, and mixed, corresponding to three different stages of reinforcement learning. The expert stage represents the highest quality trajectories, followed by the medium stage, with the mixed stage having the lowest quality. The proportions of these three categories in the dataset are 3:4:3, respectively. **Additionally, we have visualized the returns distribution of the Atari dataset using human-normalized scores for better clarity. For details, please refer to Figure B.1 in the appendix of the revised version.**
> >
> > **7. Purpose of Including Atari Experiments**
> >
> > The primary goal of our experiments on Atari is to validate whether the proposed method can work with a third modality of instructions, such as episode returns, beyond language and video. It is not aimed at achieving higher reward scores.
> >
> > While natural language is the predominant labeling (instruction) modality in robotics research, trajectories can be interpreted from multiple perspectives. In addition to language, this paper introduces the Atari benchmark to explore the feasibility of constructing the latent space using behavioral quality information (e.g., high-quality behaviors achieving high cumulative rewards). Notably, an expert human player in Atari can achieve not only high scores but also deliberately low scores, showcasing the ability to “control the score.” This skill requires a higher level of intelligence than simply achieving a high score. Experiments demonstrate that our method adapts to different instruction modalities and can interact with the environment to achieve user-specified reward levels. During inference, GROOT-2 can accept either returns as instructions or an Atari video segment. The encoder infers a latent representation from the video, guiding the policy to generate trajectories with similar quality. This approach explores an area previously underexplored in the literature.
> >
> > Furthermore, conducting experiments on Atari allows us to observe the impact of weakly supervised learning algorithms on the latent space compared to self-supervised methods, as returns provide continuously varying labels. As shown in Figure 7, our comparisons highlight that weak supervision more effectively shapes the latent space according to human-labeled intent, demonstrating its superiority in aligning latent representations with user expectations.
> >
> > We believe that comparing our method with reinforcement learning baselines such as Multi-Game DT and CQL is not appropriate. On one hand, these reinforcement learning-based approaches are specifically optimized to maximize the reward function. On the other hand, Decision Transformer (DT)-like methods rely on step-wise conditional signals, whereas GROOT-2 operates on episode-wise conditional signals. Clearly, the denser step-wise conditional signals used in DT training provide more granular guidance, making it easier for the policy to output optimal actions. And, we do not aim at achieving super high reward scores.

---

> > > ### Author Response · Authors · 2024-11-23
> > > **Response to Reviewer QHCB - Part 3**
> > >
> > > **8. A New Method To Leverage Unlabeled Trajectories**
> > >
> > > Thank you very much for suggesting this approach. The proposed method involves first training a classifier (specifically, a video description model) on labeled data, then using this model to generate textual labels for all unlabeled video data, ultimately framing the problem as a classic language-conditioned policy learning task. This approach appears promising for further improving GROOT’s performance.
> > >
> > > However, training a video description model with limited text-video data to generate high-quality textual descriptions for unseen video data is a highly challenging task. We plan to explore this direction further in future work. Thank you again for your valuable suggestion.
> > >
> > >
> > >
> > > **9. Some Baselines Are Uninformative**
> > >
> > > Thank you for your suggestion. We carefully reviewed the selection of baselines and agree that using RT-2 as a baseline is not appropriate. RT-2 is pre-trained on a wide range of auxiliary tasks, such as visual question answering and inverse action prediction, leveraging significantly more data than our approach. As a result, we have decided to remove RT-2 from the Language Table and SimplerEnv benchmarks.
> > >
> > > On the other hand, RT-1 and LAVA were at least trained on the Language Table dataset, representing some of the powerful imitation learning models available. Notably, our proposed method, GROOT-2, outperforms both LAVA and RT-1 using a policy trained with only 50% of the text labels from the Language Table dataset, demonstrating the high data efficiency of GROOT-2.
> > >
> > > Thank you again for your valuable feedback, helping us refine presentation of our results.
> > >
> > >
> > > **10. How Does Scaling Up Labeled Trajectories Impact Performance**
> > >
> > > | Labeled             | 0%           | 10%          | 25%          | 40%          | 50%          | 80%          |
> > > | ------------------- | ------------ | ------------ | ------------ | ------------ | ------------ | ------------ |
> > > | Success Rate (in %) | $10_{\pm 2}$ | $34_{\pm 8}$ | $65_{\pm 5}$ | $73_{\pm 8}$ | $82_{\pm 8}$ | $83_{\pm 5}$ |
> > >
> > > To evaluate the impact of labeled trajectory proportions in the training set on the instruction-following capabilities of GROOT-2, we conducted experiments on the Language Table benchmark. The total number of trajectories remained constant across different dataset configurations, with only the proportion of trajectories containing text labels varying. Table reports the success rate achieved by GROOT-2 conditioned on language. At low labeled data proportions $(0\%-25\%)$, the success rate rapidly increased from $10\%$ to $65\%$, indicating that labeled data significantly influences model performance. However, as the labeled data proportion increased to $50\%-80\%$, the success rate plateaued, rising slightly from $82\%$ to $83\%$, demonstrating diminishing marginal gains from additional labeled data. Therefore, under resource constraints, a labeled data proportion of $50\%$ may represent the optimal balance between performance and cost.
> > >
> > > Thank you again for your valuable suggestion. We have updated the new experimental results in the revised version of the paper. Please refer to the content around Figure 9 in the main text.
> > >
> > > **11. Questions about  Error Bars**
> > >
> > > Thank you for your valuable feedback. We have added error bars to the Minecraft experiment results in Table 1 of the revised manuscript.
> > >
> > > **12. Details about \beta_1 and \beta_2**
> > >
> > > We present the hyperparameters used in our approach in Appendix Table 4, where $\beta_1 = \beta_2 = 0.1$.
> > >
> > > The choice of $\beta_1$ is critical and requires careful tuning to avoid extreme behaviors. When $\beta_1$ is too large (e.g.,$ \beta_1 \to 1$), posterior collapse occurs (indicated by an increase in $R$), and the latent variable $z$  no longer influences the policy’s behavior. Conversely, when $\beta_1$  is too small (e.g., $\beta_1 \to 0$ , resulting in a decrease in $R$), the model exhibits severe mechanical imitation. In this case, the latent variable $z$  merely encodes the robotic arm’s movement trajectory in the reference video rather than meaningful task information.
> > >
> > > On the other hand, increasing $\beta_2$  helps the latent space better capture task-relevant information. Empirically, we found that setting $\beta_1 = \beta_2 = 0.1$ strikes a good balance between these two phenomena, ensuring effective task representation while avoiding the aforementioned issues.

---

> > > > ### Author Response · Authors · 2024-11-23
> > > > **Response to Reviewer QHCB - Part 4**
> > > >
> > > > **13. Finetune on New Tasks**
> > > >
> > > > We conducted pretraining on the Language Table benchmark across four major task categories: “move block to block,” “move block to absolute location,” “move block to relative location,” and “separate two blocks.” Subsequently, we fine-tuned the model on the “move block to block relative location” task category.
> > > >
> > > > Our findings show that the behavior cloning loss was already very low at the start of fine-tuning ($\sim 390$), significantly lower than when training solely on the “move block to block relative location” category ($\sim 1000$). This demonstrates that our proposed method is highly effective for fine-tuning on a new set of tasks.
> > > >
> > > >
> > > >
> > > > **Thank you once again for providing such detailed and valuable feedback. Your insights have greatly helped us refine and improve our work. We hope that our responses and updates address your concerns. If you feel our clarifications and revisions resolve the issues, we would be deeply grateful for your consideration in improving the score. Thank you for your thoughtful review and support!**

---

> > > > > ### Comment · Reviewer_QHCB · 2024-11-27
> > > > >
> > > > > Thanks for your rebuttal. I will keep my original score.

---

### Author Response · Authors · 2024-11-23
**Summary of Revisions Made to the Paper**

**1. Added Experiments on Backbone Parameter Initialization:**

We included new experiments to analyze the impact of backbone parameter initialization on model performance.

**2. Ablation Study on Labeled Demonstrations:**

We conducted and reported an ablation study on the number of labeled demonstrations used during training.

**3.	Ablation Study on Loss Functions:**

We introduced experiments analyzing the contributions of the language loss $\mathcal{L}\_\text{lab}$ and video loss $\mathcal{L}\_\text{dem}$.

**4. Clarifications on Ambiguous Terminologies:**

We clarified potentially ambiguous terms, such as “play data,” “unlabeled,” and “number of encoders,” to improve readability and understanding.

**5. Reorganized Experiment Section with Main Questions:**

We added a Main Questions section at the beginning of the experiment section and rewrote the subheadings in question format for better structure and flow.

**6. Reported Ratios of Labeled and Unlabeled Demonstrations:**

We included details on the proportions of labeled versus unlabeled demonstrations used for each benchmark.

**7. Enhanced t-SNE Visualizations with Legends:**

We added legends to the t-SNE visualizations to make them more interpretable.

**8. Expanded Related Works Section:**

We added discussions on several new relevant works in the Related Works section.

**9. Added Distribution of Episode Returns in Atari Supplementary Materials:**

We included a new figure in the Atari section of the supplementary materials to illustrate the distribution of episode returns.

These updates aim to enhance the clarity, completeness, and overall quality of the manuscript.

---

### Meta-Review · Area_Chair_pYmr · 2024-12-18

**Metareview:**

This paper proposes an imitation learning-style method for learning from multimodal instructions in a weakly supervised setting (partially-labeled language instructions). The paper's main claim is that it presents an effective general and flexible approach for making use of the partially-labeled language instructions. The method's bases is a VAE-based BC objective, combined with an cross entropy-based 'alignment' term of two separate encoders (one for language instructions, one from the environment observations), which serves to make use of the labeled subset of the data. Experiments support the main claim: the full scope of the ablations show that the approach improves both with the addition of unlabeled demonstrations (those without language labels) and labeled demonstrations (those with language labels).

After the rebuttal period, the main remaining weakness appears to be a limited comparison to relevant architectures for this problem. I consider this weakness relevant, but it does not undermine the main claim of the paper. Therefore, I recommend acceptance.

A minor comment I have myself is the oddness of using the cross-entropy in Eq. 3 to align the encoders, instead of a KL-divergence (computable analytically due the forms of the encoders), as well as the importance of using a stop-gradient term. I recommend considering these variants.

**Additional Comments On Reviewer Discussion:**

Multiple reviewers pointed to the absence of ablations of the objective's components. The authors responded with several ablation experiments. Reviewer SgYT engaged with the authors' responses and stated some remaining concerns requesting clarifications. The authors responded with clarifications that address the clarification concerns, in my opinion.

The two reviewers who rated a 5 did not provide substantial replies to the authors' responses, despite reminders to do so and encouragement to justifying their stances. Their only replies were:

Reviewer QHCB: https://openreview.net/forum?id=S9GyQUXzee&noteId=KTFv4agTzc
> Thanks for your rebuttal. I will keep my original score.

and

Reviewer 2co2: https://openreview.net/forum?id=S9GyQUXzee&noteId=44xadCz851
> Thank you authors for the response and extra experiments. I think they partially addressed my concern. I'd like to maintain my original evaluation.

Reviewer Mcj1 (score 6) also did not justify their score: https://openreview.net/forum?id=S9GyQUXzee&noteId=KIYkLUjUjB
> Thanks for authors' detailed explanation! I'll keep my original score.

Therefore, I have estimated whether the authors sufficiently addressed the concerns of the reviewers who rated this paper with a 5.

I found it a bit difficult to follow the response to reviewer QHCB -- the numbering and labeling of the responses did not correspond to the numbering and labeling of the weaknesses stated by QHCB. In my opinion, the most important concerns from QHCB were "How much supervision?", "Missing ablation on amount of labeled data", and "No ablations/experiments describing model architectures ... Baselines are uninformative ... A clean experiment here would be retraining an open source Octo/OpenVLA model on all the labeled data only" (I've lumped together concerns 2 and 5, these are both fundamentally about baselines and architectural ablations). Of these concerns, the author response satisfactorily addressed the first 2, but not the last (a retrained open-source Octo/OpenVLA model was not used).

In my opinion, the most relevant concern of Reviewer 2co2 was questioning the novelty of the method. The authors addressed this by pointing out that the proposed training objective function is novel, and an additional ablation confirms its effectiveness.

Altogether, I think the main remaining concern was the absence of comparison to retrained versions of Octo/OpenVLA, although this concern does not undermine the paper's main claim.

---

### Decision · Program_Chairs · 2025-01-22

Accept (Poster)